# Control of non-REM sleep by ventrolateral medulla glutamatergic neurons projecting to the preoptic area

Sasa Teng[1,2], Fenghua Zhen[1,2], Li Wang[3,4], Jose Canovas Schalchli[3,4], Jane Simko [1,4,5], Xinyue Chen[1,4], Hao Jin[3,4], Christopher D. Makinson [1,4,5] & Yueqing Peng [1,2] ✉

Understanding the neural mechanisms underlying sleep state transitions is a fundamental goal of neurobiology and important for the development of new treatments for insomnia and other sleep disorders. Yet, brain circuits controlling this process remain poorly understood. Here we identify a population of sleep-active glutamatergic neurons in the ventrolateral medulla (VLM) that project to the preoptic area (POA), a prominent sleep-promoting region, in mice. Microendoscopic calcium imaging demonstrate that these VLM glutamatergic neurons display increased activity during the transitions from wakefulness to Non-Rapid Eye Movement (NREM) sleep. Chemogenetic silencing of POA-projecting VLM neurons suppresses NREM sleep, whereas chemogenetic activation of these neurons promotes NREM sleep. Moreover, we show that optogenetic activation of VLM glutamatergic neurons or their projections in the POA initiates NREM sleep in awake mice. Together, our findings uncover an excitatory brainstem-hypothalamic circuit that controls the wake-sleep transitions.

Sleep and wakefulness are actively controlled by the interplay of distinct neural circuits in the brain[1,2]. Centered on the mutual inhibition between sleep-promoting and wakefulness-promoting circuits, a flip-flop model has been proposed to explain the mechanisms of sleep-wake state transitions[3,4]. The discovery of the ascending reticular activating system[5], the orexin neurons[6,7], and further studies of neuromodulatory systems have greatly advanced our understanding of the neural circuits supporting wakefulness[2,8,9]. By contrast, the neural circuits controlling sleep processes have remained elusive. Several sleep-promoting regions have been identified, including the preoptic area (POA, particularly the ventrolateral preoptic area and median preoptic nucleus, or VLPO and MPO)[10–14], the parafacial zone (PZ)[15,16], and the basal forebrain (BF)[17]. However,

whether these brain structures are involved in sleep initiation or sleep maintenance remains largely unknown. GABAergic neurons in these brain regions are sleep-active, and activation of these neurons promotes NREM sleep[2]. For instance, Chung et al. demonstrated that tuberomammillary nucleus (TMN)-projecting GABAergic neurons in the POA were both sleep active and sleep promoting[18]. Sleep-active neurons have been also found in a subset of GABAergic cortical interneurons that produce neuronal nitric oxide synthase (nNOS)[19,20]. In addition to the role of GABAergic neurons in sleep control, a recent study identified an excitatory circuit in the perioculomotor (PIII) region of the midbrain that promotes NREM sleep[21]. They found that CALCA-expressing glutamatergic PIII neurons project to the POA and the medulla, and optogenetic activation of these projections can

[1]Institute for Genomic Medicine, Vagelos College of Physicians and Surgeons, Columbia University, New York, NY 10032, USA. [2]Department of Pathology and Cell Biology, Vagelos College of Physicians and Surgeons, Columbia University, New York, NY 10032, USA. [3]Zuckerman Mind Brain Behavior Institute, Department of Biochemistry and Molecular Biophysics, Columbia University, New York, NY 10027, USA. [4]Department of Neuroscience, Columbia University, New York, NY 10032, USA. [5]Department of Neurology, Vagelos College of Physicians and Surgeons, Columbia University, New York, NY 10032, USA. ✉e-mail: yp2249@cumc.columbia.edu

promote NREM sleep. More recently, glutamatergic neurons that promote NREM sleep have been also found in the ventrolateral periaqueductal gray (vlPAG) of the midbrain[22] and in the posterior thalamus[23].

The POA is the most intensively studied sleep-active and sleep-promoting region. Immunohistochemistry studies in rats showed the existence of sleep-active GABAergic neurons in the VLPO and a correlation between the number of c-Fos positive cells and the amount of NREM sleep[10,24]. Consistently, electrophysiological recordings demonstrated that the firing rate of VLPO sleep-active neurons correlates with the depth of NREM sleep (indicated by EEG delta power)[11,25]. Notably, VLPO sleep-active neurons displayed lower activity in the wake–sleep transition period than in the subsequent sleep episode (discharge rates further increased significantly from light to deep NREM sleep)[25]. These findings suggest that VLPO neurons might be involved in sleep maintenance whereas other neurons may be responsible for sleep initiation.

Here, we sought to identify neuronal populations that might contribute to the transitions from wakefulness to sleep (i.e., sleep initiation). Using c-Fos activity approach and retrograde labeling, we identified a population of POA-projecting glutamatergic neurons in the ventrolateral medulla that are active in response to repeated wake–sleep transitions. Using microendoscopic calcium imaging, we further demonstrated that a subset of medulla glutamatergic neurons are preferentially active during NREM sleep. We then selectively inhibited and activated these medulla neurons in mice and examined the effects of these manipulations on sleep transitions. We found that inhibition of POA-projecting medulla neurons disrupts the transitions from wakefulness to sleep. Conversely, activation of these medulla glutamatergic neurons reliably induces the wake–sleep transitions. These results demonstrate a previously unknown brainstem-hypothalamic circuit that controls wake–sleep transitions.

## Results
### Medulla glutamatergic neurons in sleep regulation
To identify neuronal populations involved in the transitions from wakefulness to sleep, we measured c-Fos activity in a sleep deprivation (SD) paradigm. In a custom-built behavioral chamber, we programmed an intermittent movement of a sweeping rod (30 s on, 90 s off) to disrupt the animals' sleep for 6 h during the light cycle (Fig. 1a, Supplementary Fig. 1). The movement pattern forced the animals to briefly wake up from NREM sleep so that it induced repeated wake–sleep transitions (Fig. 1b, Supplementary Fig. 1e). To control for any effects of increased motor activity, we programmed a continuous sweeping movement for 2 h before perfusion in another group of animals. In a third group, the animals were allowed a 2-h period of recovery sleep (RS) following deprivation, which significantly reduced wake–sleep transitions (Supplementary Fig. 1e). Then, we examined elevated c-Fos+ brain regions in SD mice, compared to control and RS groups. Early transection studies suggest the existence of sleep-promoting neurons in the medullary brainstem[26]. Indeed, among c-Fos+ brain regions, we identified a population of cells in the ventrolateral medulla (VLM), mostly in the lateral paragigantocellular nucleus (LPGi) and its surrounding area (Fig. 1c). By contrast, we observed fewer c-Fos+ cells in the control and RS animals. To examine the identity of these c-Fos+ cells, we performed triple-staining of *Fos*, the glutamatergic marker *Slc17a6* (encoding vGLUT2) and the GABAergic marker *Slc32a1* (encoding vGAT) using fluorescence in situ hybridization (FISH) in SD mice. Our data demonstrated that the majority of c-Fos+ cells (91.7 ± 1.8%, SEM) are glutamatergic, but not GABAergic (Fig. 1d, Supplementary Fig. 2).

To study the activity of VLM glutamatergic neurons in wake/sleep cycles, we then performed in vivo recording by using microendoscopic calcium imaging technique[27,28]. We stereotaxically injected AAV-FLEX-GCaMP6f into the VLM of Vglut2-Cre mice and implanted a gradient refractive index (GRIN) lens (Inscopix, 0.5-mm diameter) above the injection site (Fig. 2a). We found that imaging the brainstem in freely moving animals typically involves significant movement artifacts. To overcome this issue, we glued two anchoring tungsten wires to the GRIN lens to stabilize the field of view as previously shown[29]. Three weeks after recovery, we imaged VLM glutamatergic neurons while the animals experienced natural wake–sleep cycles. During each imaging session, brain states were classified based on EEG and EMG recordings. We observed high spontaneous activity in subsets of the VLM cells during sleep (Fig. 2b). To quantifying the relative activity of each neuron in different brain states, we plotted its NREM-wake (N-W) selectivity index (see Methods for details) versus the REM-wake (R-W) selectivity index. We found that the majority of VLM glutamatergic cells (57% among 70 cells) were selectively active during wakefulness, particularly during periods with higher motor activity, indicated by higher EMG amplitudes and higher body acceleration (Supplementary Fig. 3). This finding suggests a functional role of these neurons in motor control. Indeed, a previous study showed that glutamatergic neurons in the LPGi are involved in locomotion[30]. Importantly, we identified a population of VLM glutamatergic neurons (16%) that

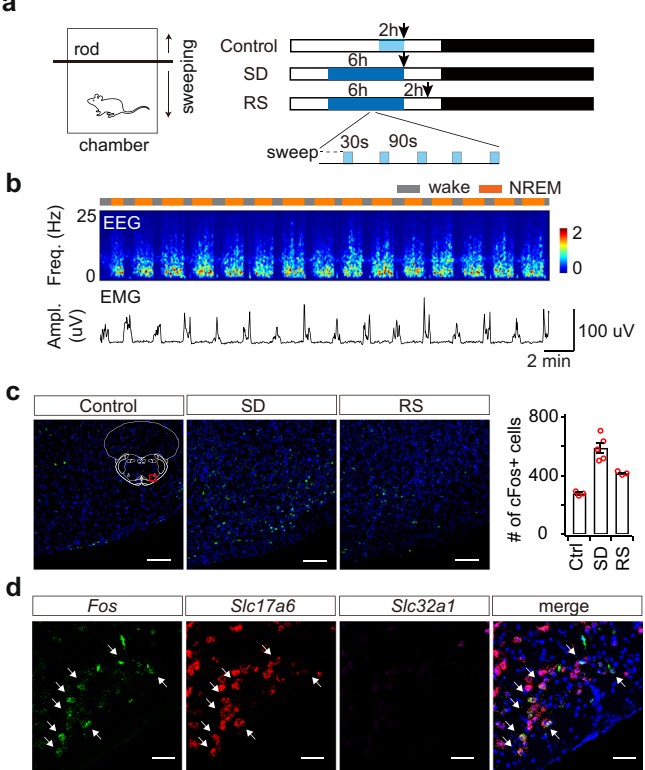

**Fig. 1 | Medulla glutamatergic neurons are active during wake–sleep transitions. a** Schematic of sleep deprivation. Animals were divided into three groups: 1) control, 2-h sweeping before perfusion, 2) sleep deprivation (SD) by 6-h intermittent sweeping (30 s on, 90 s off), 3) recovery sleep (RS) after deprivation. Arrows indicate perfusion time. **b** Representative EEG spectrogram and EMG trace showing 30-min sleep pattern at the 5th hour of sleep deprivation. Brain states: gray, wake; orange, NREM sleep. **c** Fluorescence images of c-Fos immunostaining (green) in the VLM of three groups: control, SD, and RS. Blue, DAPI. Scale bars, 100 μm. Inset, mouse brain coronal section (Bregma −6.9 mm), adapted from Allen mouse brain atlas (http://atlas.brain-map.org). Red box indicated the region of fluorescence images. Right, Quantitation of c-Fos+ cells in the VLM in each group (*n* = 3 mice for control, *n* = 5 for SD, *n* = 3 for RS). Data are presented as mean ± SEM. Source data are provided as a Source Data file. **d** Triple fluorescence in situ hybridization (FISH) of *Fos* and neuronal markers in the VLM of a SD mouse (*n* = 2 mice). *Slc17a6*, vesicular glutamate transporter 2; *Slc32a1*, vesicular GABA transporter. Arrows indicated the overlapped cells between *Fos* and *Slc17a6*. Blue, DAPI. Scale bars, 50 μm.

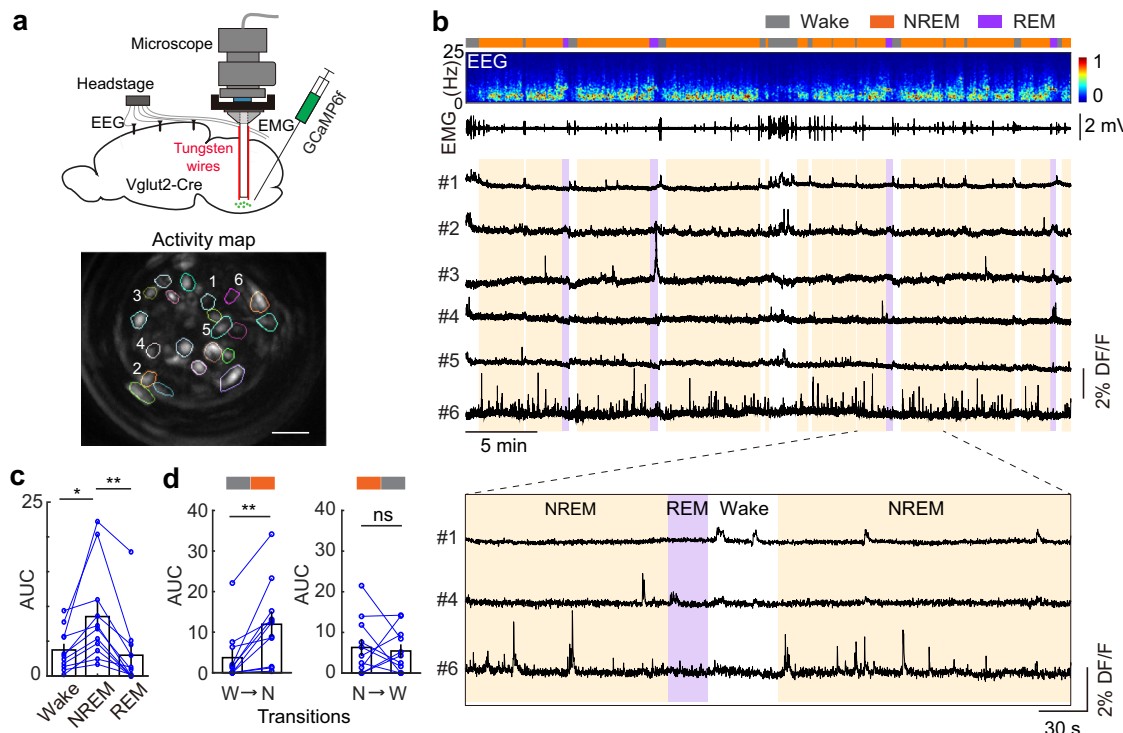

**Fig. 2 | Imaging medulla glutamatergic neurons during sleep. a** Schematic of microendoscopic calcium imaging in the VLM. Tungsten wires attached to the GRIN lens were used to stabilize the field of view. Bottom, activity map of an example imaging session in a Vglut2-Cre mouse. Scale bar, 50 μm. **b** An example showing brain states (gray for wake, orange for NREM sleep, purple for REM sleep), EEG power spectrogram (0–25 Hz), EMG, and representative calcium traces (DF/F) in an imaging session. Brown shadows indicate NREM sleep. Purple shadows indicate REM sleep. Bottom, enlarged window showing calcium activity in 3 cells (#6 as a NREM sleep active cell) during wake–sleep transition. **c** Quantification of calcium activity of NREM sleep active cells ($n = 11$ neurons from 4 mice, *$P = 0.011$ between wake and NREM, **$P = 0.0007$ between NREM and REM, $P = 0.621$ between wake and REM, two-sided paired $t$-test) in each brain state. Each line represents one neuron. Bar graph, average across neurons. Error bars, SEM. **d** Left, quantification of calcium activity during the transitions from wakefulness (W) to NREM sleep (N) in the NREM sleep active cells in panel **c**. Right, quantification of calcium activity during the NREM-to-wake transitions in the NREM active cells ($n = 11$ neurons from 4 mice, two-sided paired $t$-test, **$P = 0.002$ for W–N, $P = 0.716$ for N–W, n.s. no significance). Error bars, SEM. Source data are provided as a Source Data file.

displayed the highest activity during NREM sleep, compared to that in wake and REM sleep (Fig. 2c). These NREM sleep active cells have not been reported previously. Further analysis demonstrated that these neurons displayed increased activity during the transitions from wakefulness to NREM sleep whereas their activity remains almost unchanged during the NREM-to-wake transitions (Fig. 2d). These data suggest that NREM active neurons in the VLM might play an active role in the wake-to-sleep transitions. Another population of cells (27%) displayed the highest activity during REM sleep (Supplementary Fig. 3d). Together, these results indicate that the VLM contains functionally heterogenous populations of glutamatergic neurons: sleep-active neurons mixed with motor neurons. Importantly, our data reveal a population of NREM sleep active glutamatergic neurons in the VLM.

## Medulla glutamatergic neurons project to the preoptic area

We reasoned that sleep-promoting neurons should project to other sleep centers to orchestrate sleep behavior. To test this, we performed retrograde tracing experiments by injecting retrograde virus carrying Cre-recombinase (AAVrg-Cre) in the POA in a reporter line (Ai9, Fig. 3a). Four weeks after viral injection, we perfused the mice and processed the whole brains by clear, unobstructed brain imaging cocktails and computational analysis (CUBIC) clearing and 3D imaging with light-sheet fluorescent microscopy[31]. This technique uncovered multiple brain regions that project to the POA, with the majority in the forebrain (Fig. 3b). Among tdTomato-labeled cells in the brainstem, we identified a population of neurons in the similar VLM region where c-Fos+ cells in SD experiments were located (Fig. 3b, c). The labeled cells

were mostly clustered in areas overlapping C1 adrenergic and A1 noradrenergic fields in the LPGi (Supplementary Fig. 4)[32]. However, only a small portion ($18.08 \pm 4.11\%$) of retrogradely labeled cells are tyrosine hydroxylase (TH) positive based on FISH experiments (Supplementary Fig. 5a, b). To examine the relationship between c-Fos+ cells and POA-projecting neurons in the VLM, we then performed c-Fos staining and retrograde labeling in sleep-deprived Ai9 transgenic mice. Our data demonstrated that half ($50.1 \pm 1.8\%$, SEM) of POA-projecting VLM neurons are c-Fos+ (Fig. 3d). In addition, we didn't observe significant overlap between TH and c-Fos in the VLM (Supplementary Fig. 5c), suggesting that TH + VLM cells are not activated by the SD treatment. Based on these results and the Fos FISH data (Fig. 1d), we reasoned that POA-projecting VLM neurons are glutamatergic. To confirm it, we performed FISH and retrograde labeling in Ai9 mice. Indeed, we found that the majority ($74.0 \pm 3.47\%$, SEM) of the retrogradely labeled cells (tdTomato+) in the medulla were co-labeled by *Slc17a6* in the medulla (Supplementary Fig. 5d). By contrast, we observed almost no overlap between tdTomato and *Slc32a1* ($0.23 \pm 0.23\%$, SEM). Together, these results indicated that VLM glutamatergic neurons, but not GABAergic neurons project to the POA.

The POA contains different cell types of neurons. For instance, about 85% of the neurons in the VLPO contain the inhibitory neurotransmitters galanin and GABA[10,33], which are sleep-active and sleep-promoting[10–14]. To test whether VLM neurons directly synapse with POA GABAergic neurons, we used a Cre-dependent monosynaptic retrograde viral reporter to study their connections[34,35]. In GAD2-Cre mice, we injected AAV-FLEX-G(N2C)-mKate[35] and AAV-FLEX-TVA-mCherry into the POA. Two weeks after AAV injection, we injected

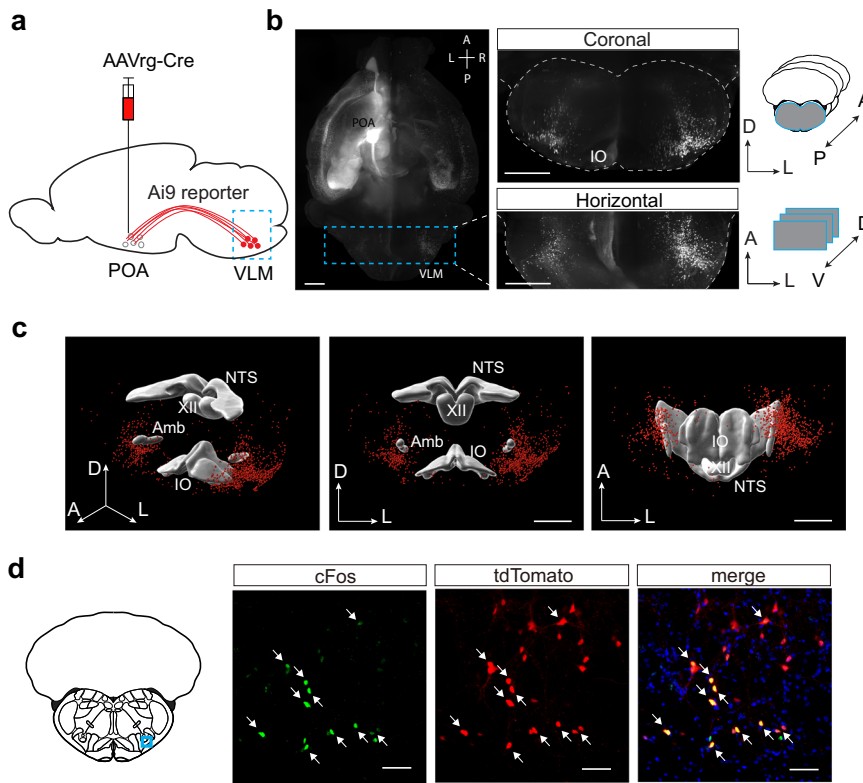

**Fig. 3 | Medulla glutamatergic neurons project to the preoptic area. a** Schematic illustrating retrograde labeling experiments in Ai9 reporter mice injected with AAVrg-hSyn.Cre.WPRE.hGH in the preoptic area (POA). VLM, ventrolateral medulla. **b** Left, maximum-intensity z stack of POA-projecting neurons in the whole mouse brain, cleared with CUBIC and imaged with light-sheet fluorescent microscopy (a representative example from 3 mice). The dashed-line box illustrating the anterior and posterior boundaries used for reconstruction of coronal and horizontal views on the right. Right top, Maximum-intensity z stack of POA-projecting neurons in the optically sliced coronal sections of the brainstem. Right bottom, Maximum-intensity z stack of POA-projecting neurons in the optically sliced horizontal sections of the brainstem. Mouse brain figures in **b** and **d** adapted from Allen mouse brain atlas. Scale bars, 1 mm. A, anterior; P, posterior; D, dorsal; V, ventral; L, lateral. **c** Three-dimensional reconstruction of POA-projecting medulla neurons in the perspective, front, and ventral view, respectively (from Bregma −6.0 mm to −8.0 mm). Scale bars, 1 mm. **d** Fluorescence images of cFos immunostaining (green) and tdTomato expression (red) in the VLM (blue box on the coronal section) of a sleep-deprived Ai9 mouse injected with AAVrg-Cre in the POA ($n = 2$ mice). Arrows indicated the co-stained cells of cFos and tdTomato. Blue, DAPI. Scale bars, 50 μm. NTS nucleus of the solitary tract, XII hypoglossal nucleus, Amb nucleus ambiguus, IO inferior olivary complex.

RABV-ΔG-GFP-EnvA[35] into the same POA region. Seven to ten days after rabies injection, mice were perfused to examine GFP expression in the brains. As expected, we identified GFP-labeled neurons in the VLM (Fig. 4a), indicating that VLM neurons are anatomically connected with GABAergic neurons in the POA. To demonstrate the functional connectivity between VLM and POA, we performed two sets of experiments. Firstly, we performed fiber photometry recording in GAD2-Cre mice. We injected AAV-CaMKIIa-ChR2, which predominantly targets glutamatergic neurons (Supplementary Fig. 6a) into the VLM and injected AAV-FLEX-GCaMP6s into the POA. After two weeks, we optogenetically activated VLM neurons and simultaneously recorded photometric signals in the POA. We observed robust increases in calcium signal in POA GABAergic neurons upon light stimulation, indicating the existence of an excitatory connection between VLM and POA (Supplementary Fig. 6b, c). Secondly, to further study the synaptic transmission between VLM neurons and its downstream targets, we performed whole-cell recordings in acute slices of the POA while optogenetically activating ChR2-expressing axonal terminals of VLM glutamatergic neurons (Fig. 4b). In Vglut2-Cre mice, we injected AAV-DIO-ChR2 in the VLM and AAV-mDlx-mRuby2 (targeting GABAergic neurons[36]) in the POA. Upon ChR2 activation, we observed reliable excitatory postsynaptic currents (EPSCs) or potentials (EPSPs) in GABAergic neurons in the POA, which was blocked by the AMPA- and NMDA-type glutamate receptor antagonist CNQX and DL-AP5 (Fig. 4c, d, and Supplementary Fig. 7). Furthermore, consistent with rabies tracing results, we observed many short latency synaptic responses

following optogenetic stimulation of VLM axons which confirms the existence of a monosynaptic connection between VLM and POA (Fig. 4c and Supplementary Fig. 7b). Together, these results confirm that VLM glutamatergic neurons send excitatory input to GABAergic neurons in the POA.

## POA-projecting medulla neurons are required for wake–sleep transitions

Next, we conducted both loss-of-function and gain-of-function experiments to determine the role of the VLM-POA circuitry in sleep behavior. To test the necessity, we chemogenetically silenced VLM neurons and then examined wake–sleep transitions. As demonstrated by our calcium imaging data (Fig. 2), VLM glutamatergic neurons are heterogenous. To selectively target POA-projecting glutamatergic neurons in the VLM, we bilaterally injected retrograde AAV expressing Cre recombinase (AAVrg-Cre) into the POA, and bilaterally injected AAV carrying Cre-dependent inhibitory DREADDs (AAV-DIO-hM4D(Gi)) into the VLM (Fig. 5a). After two weeks of recovery, we treated animals with clozapine-N-oxide (CNO, 1 mg/kg) or saline and examined their effects on sleep behavior (Fig. 5b). We reasoned that if POA-projecting VLM neurons are required for the transitions from wakefulness to sleep, we should observe less NREM sleep and more wake time after CNO treatment. Since NREM sleep is often followed by REM sleep, we also predicted that less NREM sleep could lead to less REM sleep. Indeed, silencing POA-projecting VLM neurons significantly promoted wakefulness and suppressed both NREM sleep and REM

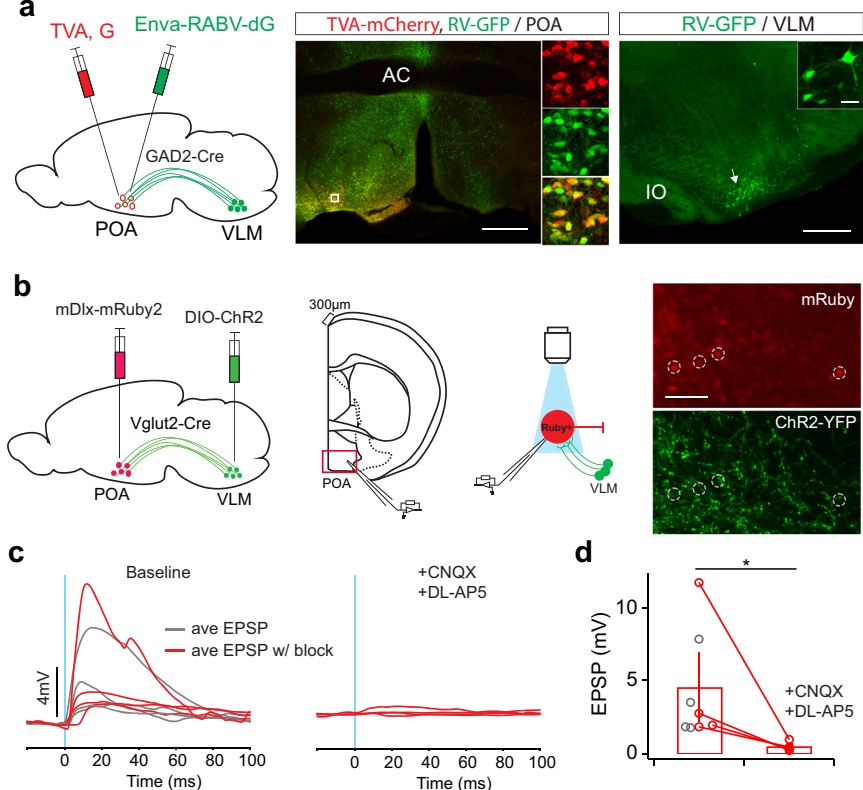

**Fig. 4 | VLM glutamatergic neurons activate GABAergic neurons in the POA.**
**a** Monosynaptic retrograde tracing from the POA to VLM in GAD2-Cre mice. Left, schematic of experimental design. The POA of GAD2-Cre animals were infected with AAV-FLEX-G(N2C)-mKate and AAV-FLEX-TVA-mCherry, followed by infection with RABV-ΔG-GFP-EnvA. Middle, fluorescence image of the injection site in the POA. Scale bar, 500 μm. Enlarged view of the region in the white box showing starter cells labeled by mCherry (red), Rabies-labeled cells labeled by GFP (green). Right, Rabies-labeled cells in the VLM indicated by GFP expression (green). Representative of 2 mice. Scale bar, 500 μm. Arrow indicates cells in inset. Scale bar, 20 μm. **b** Schematic of experimental design. AAV-DIO-ChR2-eYFP was injected unilaterally in the VLM, AAV-mDlx-mRuby2 injected contralaterally in the POA of Vglut2-Cre mice. Recordings were performed in mRuby+ cells in the POA while light

stimulation was delivered to activate ChR2-expressing VLM glutamatergic axons. Right, Fluorescence images showing expression of mRuby and ChR2-eYFP in the POA (n = 6 mice). Scale bar, 50 μm. **c** Excitatory post-synaptic potentials (EPSPs) evoked by optogenetic stimulation (blue line, 1 ms) before (left, or baseline) and after (right) glutamatergic receptor antagonists CNXQ (10 μM) and DL-AP5 (10 μM). Each trace represents averaged response across 5 sweeps in one cell. Cells with pharmacological block are highlighted in red. **d** Quantitation of average EPSP amplitudes (n = 8 cells from 6 Vglut2-Cre mice, among them, 4 cells with pharmacological block are used for the bar graph and statistical comparison, Error bars, SEM, *P = 0.028 Mann–Whitney U-test). Source data are provided as a Source Data file.

sleep (Fig. 5c–e, Supplementary Fig. 8a–c). A similar trend was also observed with a lower dose of CNO treatments (0.5 mg/kg, Supplementary Fig. 8d). Further analysis demonstrated that decreased NREM and REM sleep time were caused by decreased bout numbers of NREM and REM sleep, while the bout durations of NREM and REM sleep remain unchanged (Fig. 5d, e). By contrast, the increased wake time was caused by increased wake bout durations (Fig. 5c). In control experiments, CNO treatment (1 mg/kg) has no significant effect on sleep behavior in the wild type mice without DREADD expression (Supplementary Fig. 8e). Together, these results indicate that POA-projecting VLM neurons are an important mediator of wakefulness to NREM sleep transitions.

**POA-projecting medulla neurons are sufficient to promote NREM sleep**
We then examined the sufficiency of POA-projecting VLM neurons in wake–sleep transitions by conducting gain-of-function experiments. Using a similar viral injection strategy, we chemogenetically activated POA-projecting VLM neurons and examined its effect on sleep behavior (Fig. 6a, b). We bilaterally injected AAVrg-Cre into the POA, and unilaterally injected AAV carrying Cre-dependent excitatory DREADDs (AAV-DIO-hM3D(Gq)) into the VLM. After two weeks of recovery, we treated the animals with CNO (1 mg/kg) and saline while recording

sleep behavior. We found that chemogenetic activation of POA-projecting VLM neurons significantly increased the NREM sleep time, accompanied by a decrease of wake time (Fig. 6c–e). Further analysis showed that the increase of NREM sleep time was due to the increase of NREM sleep bouts, but not the durations of NREM sleep bouts (Fig. 6d). By contrast, the decrease of the wake time was largely due to the decrease of wake bout durations (Fig. 6c), suggesting that the animals were more likely to switch from wakefulness to sleep in response to chemogenetic activation. In addition, we observed no significant change of REM sleep (the total time, REM sleep bouts and bout durations, Fig. 6e) after CNO treatment. These results indicate that activation of POA-projecting neurons in the VLM is sufficient to promote NREM sleep.

**Medulla glutamatergic neurons control wake–sleep transitions**
The transitions from wakefulness to NREM sleep occur on the time scale of seconds. Due to the slow process of drug administration, chemogenetic manipulation lacks the temporal resolution to track each wake–sleep transition. To address this problem, we then used optogenetics to directly activate the VLM glutamatergic neurons and examine behavioral output. We stereotaxically injected AAV-DIO-ChR2-eYFP into the VLM of Vglut2-Cre mice and implanted an optical fiber above the injection site (Fig. 7a). Strikingly, we observed rapid

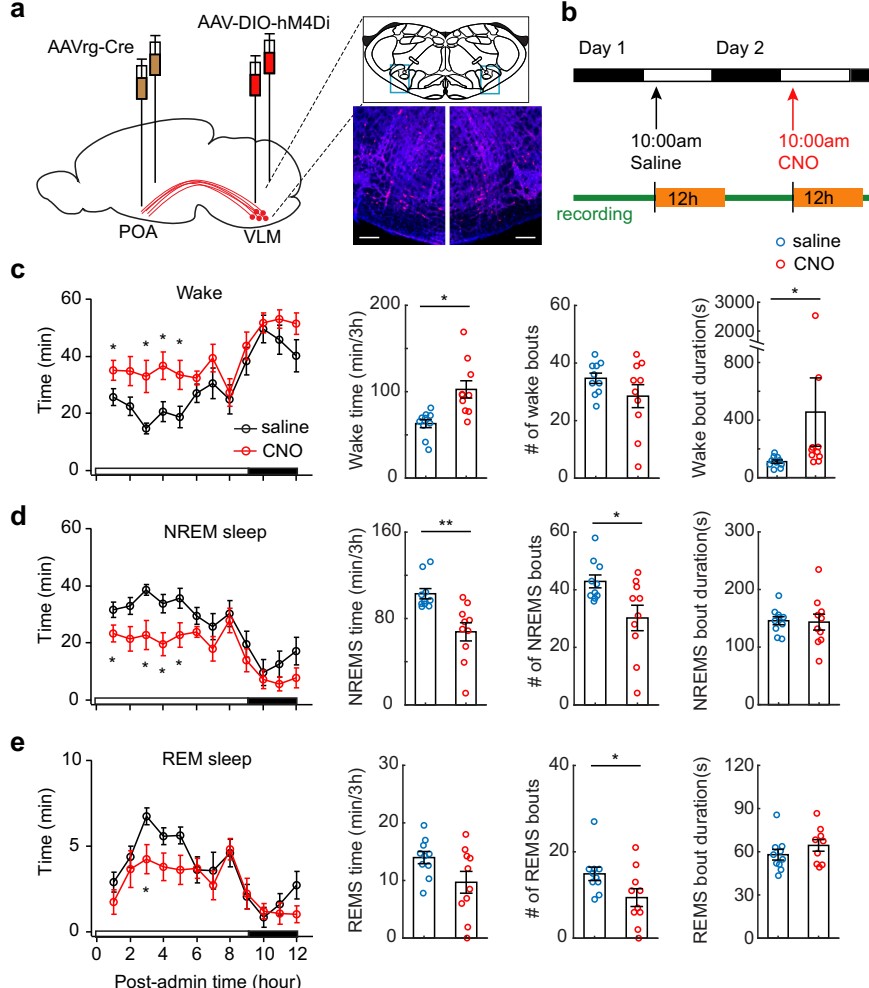

**Fig. 5 | Silencing POA-projecting medulla neurons disrupts the transitions from wakefulness to sleep. a** Schematic of chemogenetic inhibition. AAVrg-Cre was bilaterally injected in the POA and AAV9-hSyn-DIO-hM4D(Gi)-mCherry was bilaterally injected in the VLM of wild-type mice. Right, Fluorescence images of coronal sections in the VLM (blue boxes above) showing bilateral expression of inhibitory DREADDs ($n = 8$ mice). Mouse brain figure adapted from Allen mouse brain atlas. Red, mCherry, Blue, DAPI. Scale bars, 100 μm. **b** Schematic of drug administration and sleep recording. **c** Wake time per hour over a 12-h time window following CNO (1 mg/kg, i.p. red circles) and saline (black circles) treatments. Right, quantitation of the total wake time ($P = 0.011$), wake bouts ($P = 0.150$), bout durations ($P = 0.016$ without two outliers) over 3-h after treatments. **d** Quantitation of hourly NREM sleep time over 12-h, the total NREM time ($P = 0.0098$), bouts ($P = 0.023$), and bout durations ($P = 0.869$) over 3-h after CNO and saline treatments. **e** Quantitation of hourly REM sleep time over 12-h, the total REM time ($P = 0.061$), bouts ($P = 0.0198$), and bout durations ($P = 0.129$) over 3-h after CNO and saline treatments. Data are mean ± SEM in panel **c**–**e**. For all statistics, $n = 10$ animals, two-sided paired *t*-test, *$P < 0.05$, **$p < 0.01$. Source data are provided as a Source Data file.

transitions from wakefulness to NREM sleep upon the offset of laser stimulation (Fig. 7b, also see Video 1). To quantitatively examine the probability of optogenetically induced transitions from wakefulness to sleep, we applied video-based closed-loop optogenetic stimulation in awake animals. An IR-camera was used to track the animal's movement in real-time and to trigger laser stimulation automatically by wakefulness. We found that optogenetic activation of VLM glutamatergic neurons reliably initiated NREM sleep in awake mice with a transition probability of ~80% (Fig. 7c, f). Notably, optogenetic-induced NREM sleep episodes lasted a few minutes, with a distribution of bout durations similar to that in natural sleep (Fig. 7e). Moreover, sleep depth indicated by the delta power showed no significant difference between optogenetic-induced and natural NREM sleep episodes (Fig. 7e). To our knowledge, this is the first study to demonstrate that optogenetic stimulation can initiate long-lasting naturalistic NREM sleep. In Vglut2-Cre mice, we performed two control experiments: 1) no light

stimulation in ChR2-expressing mice, 2) light stimulation in YFP-expressing mice (AAV-DIO-eYFP). As expected, we observed low spontaneous but not significant wake–sleep transitions following photostimulation (Fig. 7d, f). As another control, we performed similar optogenetic experiments in GAD2-Cre mice. No significant transitions from wakefulness to NREM sleep were observed upon activation of VLM GABAergic neurons in awake animals (Fig. 7f, Supplementary Fig. 9a). Interestingly, we also observed reliable optogenetic-induced wake–sleep transitions in wildtype mice (C57BL/6J) injected with AAV-CaMKIIa-ChR2-eYFP (Fig. 7f, Supplementary Fig. 10). As shown in Supplementary Fig. 6a, this viral strategy predominantly labels glutamatergic cells in the VLM. Together, these results indicate that optogenetic activation of VLM glutamatergic neurons can reliably initiate the transition from wakefulness to NREM sleep.

A notable phenomenon observed in this study was that NREM sleep occurred after the termination of laser stimulation rather

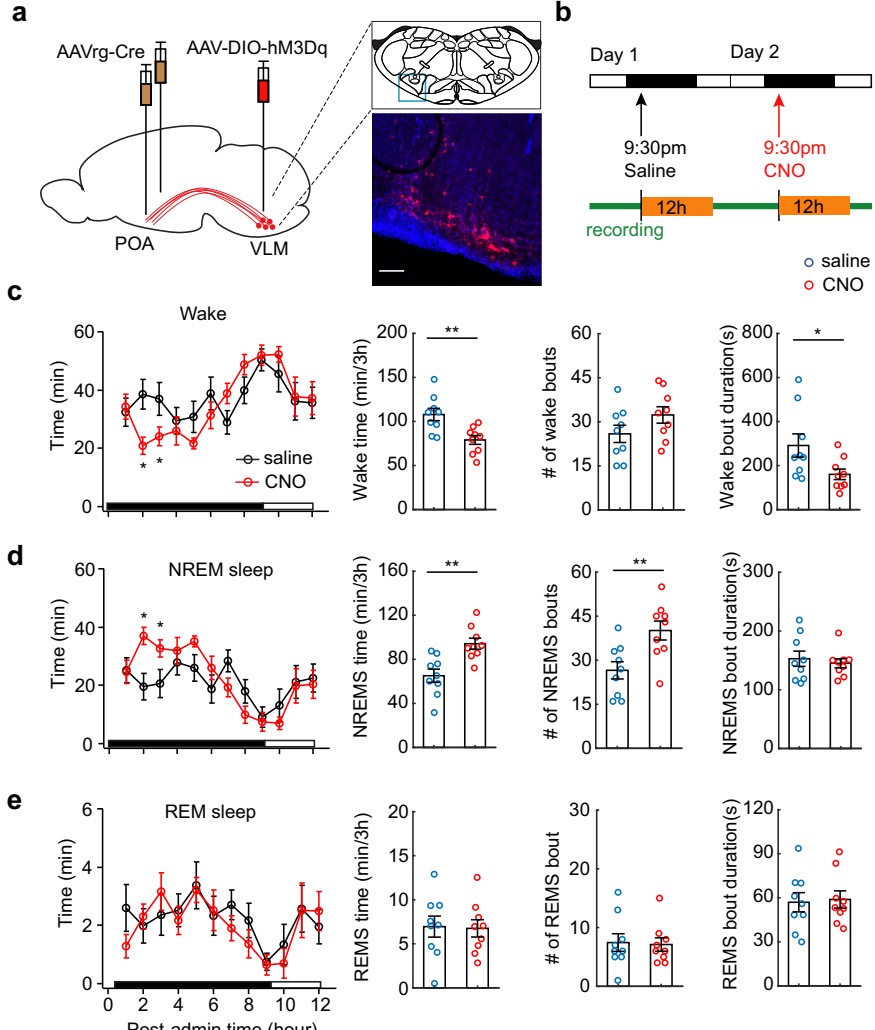

**Fig. 6 | Chemogenetic activation of POA-projecting medulla neurons promotes NREM sleep. a** Schematic of chemogenetic activation. AAVrg-Cre was bilaterally injected in the POA, AAV-DIO-hM3D(Gq)-mCherry was unilaterally injected in the VLM of wild-type mice. Right, Fluorescence image of a coronal section in the VLM (blue box above) showing expression of excitatory DREADDs (*n* = 8 mice). Mouse brain figure adapted from Allen mouse brain atlas. Red, mCherry, Blue, DAPI. Scale bar 100 μm. **b** Schematic of drug administration and sleep recording. **c** Wake time per hour over a 12-h time window following CNO (1 mg/kg, i.p. red circles) and saline (black circles) treatments. Right, quantitation of the total wake time

(*P* = 0.001), wake bouts (*P* = 0.106), bout durations (*P* = 0.032) over 3-h after treatments. **d** Quantitation of hourly NREM sleep time over 12-h, the total NREM time (*P* = 0.0008), bouts (*P* = 0.0017), and bout durations (*P* = 0.517) over 3-h after CNO and saline treatments. **e** Quantitation of hourly REM sleep time over 12-h, the total REM time (*P* = 0.861), bouts (*P* = 0.837), and bout durations (*P* = 0.797) over 3-h after CNO and saline treatments. Data are mean ± SEM in panel **c**–**e**. For all statistics, *n* = 9 animals, two-sided paired *t*-test, **P* < 0.05, ***p* < 0.01. Source data are provided as a Source Data file.

than the onset. We proposed two possibilities to explain this observation: (1) several minutes of laser stimulation may be required to induce wake–sleep transitions, and therefore the timing of observation is a coincidence due to the stimulation protocol we used; (2) nonspecific activation prevents the animals from falling asleep during the stimulation period. To distinguish between these possibilities, we first activated the VLM glutamatergic neurons for different durations (30 s and 1 min). Similarly, we found that wake–sleep transitions occurred following the offset of laser stimulation irrespective of activation durations (Supplementary Fig. 11). Shorter durations of stimulation induced slightly less wake-sleep transitions (Supplementary Fig. 11c). To test the second possibility, we used the dual viral injection strategy (see Figs. 5 and 6) to retrogradely label subsets of VLM neurons in wildtype mice. Strikingly, we found that optogenetic activation of

POA-projecting VLM neurons induced NREM sleep in awake animals following the onset of light stimulation (Fig. 7g). Together, these results excluded the first possibility and supported that nonspecific activation might be involved in offset phenomenon observed in Vglut2-Cre mice.

As reported previously[14], high-frequency light stimulation of neurons with slow firing rates can cause depolarization block, which results in acute silencing of ChR2-expressing neurons. To exclude this possibility, we performed optogenetic experiments with lower frequency stimulation (1, 2, 5, 10 Hz). We observed optogenetic-induced wake–sleep transitions in a frequency-dependent manner, with higher frequency stimulation resulting in higher transition probability (Fig. 7h).

Next, we examined the effect of optogenetic stimulation in sleeping mice by using a fixed-interval laser stimulation. In trials with

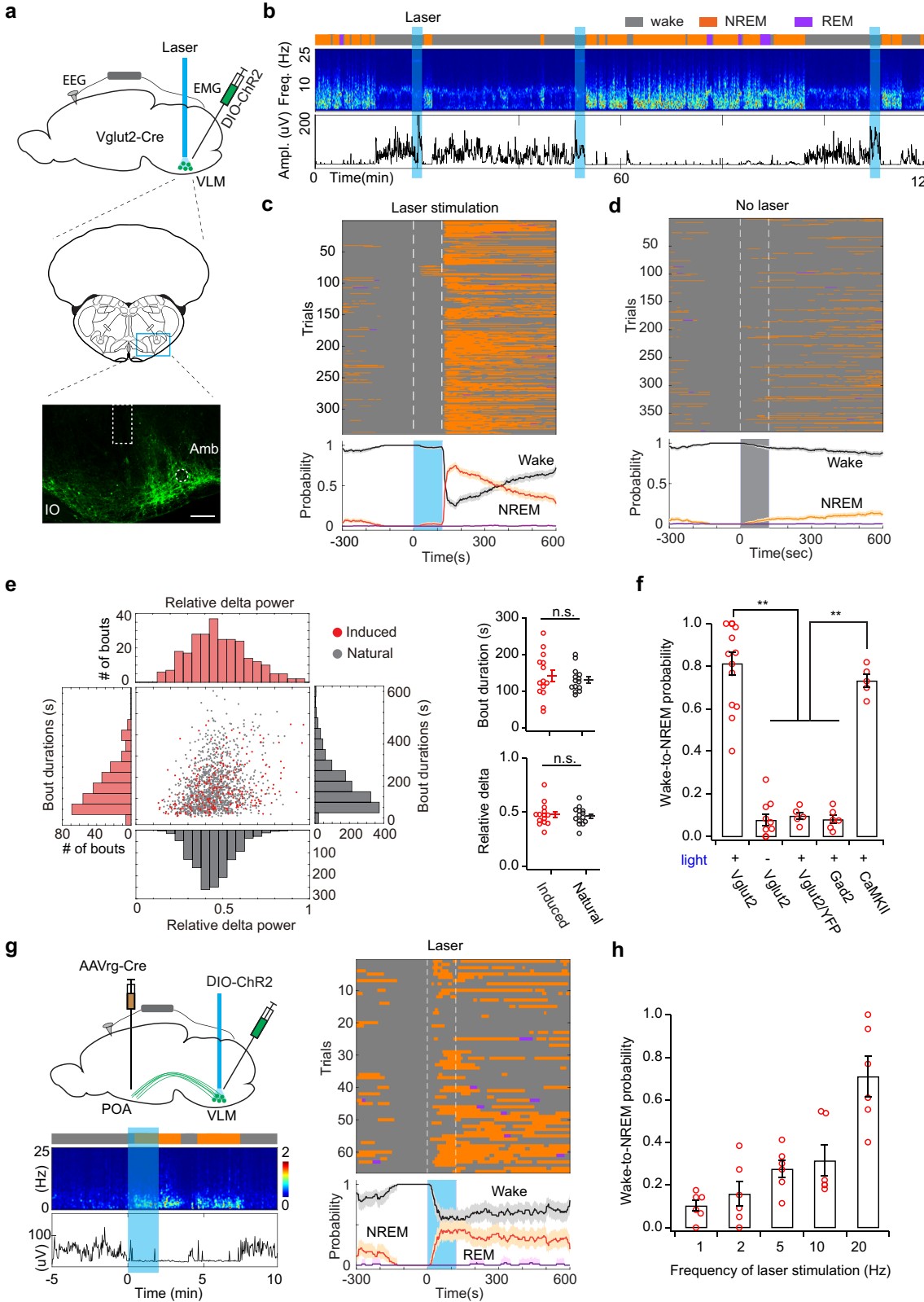

sleep before laser stimulation, we found that activation of VLM glutamatergic neurons strongly interrupted the animal's sleep and triggered immediate locomotor activity (Supplementary Fig. 12a). In most of these trials, the animals were able to switch back to NREM sleep when the laser was turned off. Notably, optogenetic-induced locomotor activity was also observed in awake animals. We speculated that the locomotion behavior might be caused by non-specific stimulation

in motor neurons in the LPGi. Capelli et. al. showed that optogenetic activation of LPGi glutamatergic neurons elicits full-body locomotion[30]. Our calcium imaging results also demonstrated motor-related neurons in the region. This optogenetic-induced nonspecific locomotion might awaken the animals from sleep. In addition, this locomotion effect might also cause disturbances and prevent animals from falling asleep during the photostimulation period (Fig. 7c).

**Fig. 7 | Optogenetic activation of medulla glutamatergic neurons induce NREM sleep in awake animals. a** Schematic of optogenetic experiment. Middle, a coronal section (adapted from Allen mouse brain atlas). Bottom, fluorescence image of VLM (blue box) in a Vglut2-Cre mouse injected with AAV-DIO-ChR2-eYFP ($n = 13$ mice). Dashed box indicates the placement of optical fiber. Amb nucleus ambiguus; IO inferior olivary complex. Scale bar 200 μm. **b** Representative EEG spectrogram and EMG trace in a 2-h session with photostimulation (20 Hz, 2 min, blue shades). Brain states: orange, NREM; purple, REM; gray, wake. **c** Brain states before, during and after laser stimulation (white dash lines) in awake animals ($n = 14$). Bottom, probability of wake (black), NREM (orange), and REM sleep (purple). **d** Brain states in no light animals ($n = 10$). **e** Scatter plots and distributions of bout durations and relative delta power during NREM sleep ($n = 14$ animals). Right, quantitation of bout durations (upper, $P = 0.588$) and relative delta power (bottom, $P = 0.703$) ($n = 14$, n.s. no significance, two-sided unpaired $t$-test). Error bars, SEM. **f** Quantification of

transition probability from wakefulness to NREM sleep induced by optogenetic activation ($n = 14$ animals for Vglut2-Cre with light, $n = 10$ for Vglut2-Cre without light, $n = 5$ for Vglut2-Cre/YFP, $n = 6$ for Gad2-Cre, $n = 5$ for CaMKII, Mann–Whitney $U$-test, **$P < 0.0006$ between Vglut2+light and control, $P < 0.008$ between CaMKII and control, control refers to Vglut2 no light, Vglut2/YFP, and Gad2+light, $P = 0.286$ between Vglut2+light and CaMKII). Error bars, SEM. **g** Optogenetic stimulation of POA-projecting VLM neurons. Left upper, schematic of experimental design. AAVrg-Cre injected unilaterally in POA, AAV-DIO-ChR2-eYFP injected in VLM in wild-type mice. Left bottom, a trial of optogenetic activation (20 Hz, 2 min). Right, brain states and probability before, during, and after laser stimulation (66 trials from 3 mice). Traces in **c**, **d**, **g** are presented as mean ± 95% bootstrap confidence intervals. **h** Quantitation of wake–sleep transition probability under different frequencies of laser stimulation (2 min, $n = 6$ Vglut2-Cre mice). Error bars, SEM. Source data are provided as a Source Data file.

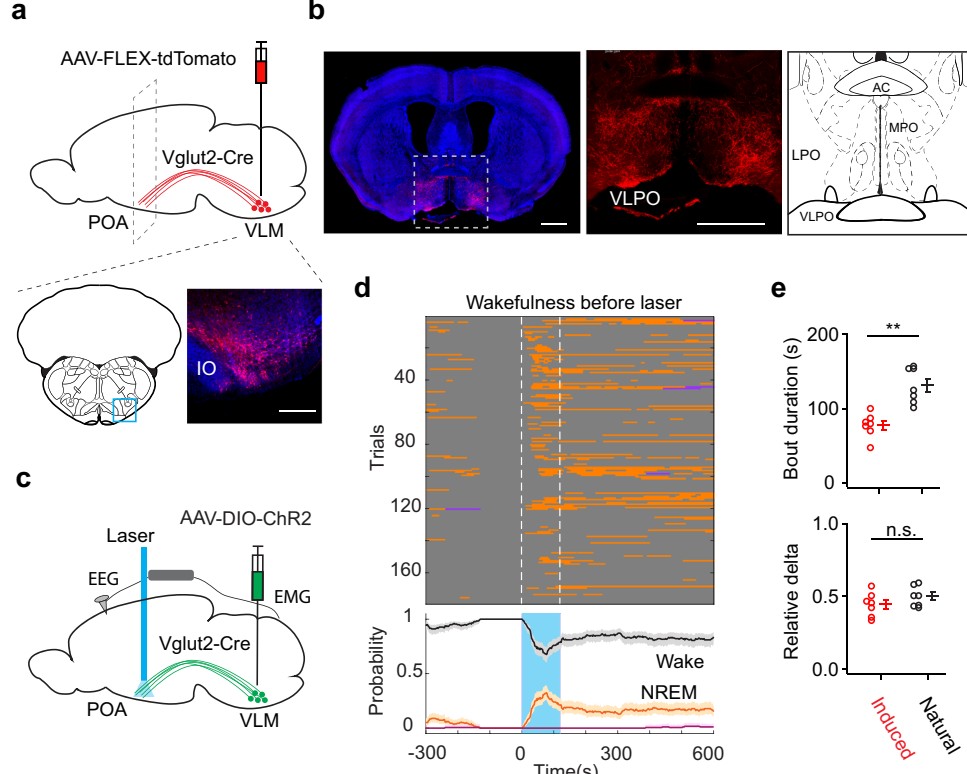

**Fig. 8 | Optogenetic activation of VLM glutamatergic terminals in the POA promotes the transition from wakefulness to NREM sleep. a** Schematic of anterograde tracing experiment. AAV1-CAG-FLEX-tdTomato (red) was unilaterally injected in the VLM (bottom) of Vglut2-Cre mice ($n = 5$ mice). Mouse brain figure adapted from Allen mouse brain atlas. Blue DAPI. IO inferior olivary complex. Scale bar, 200 μm. **b** Fluorescence image of a coronal section (Bregma 0) illustrating axonal terminals (red) in the POA (enlarged in the middle, 5 mice in total). Blue DAPI. Scale bar, 1 mm. Right, diagram of the POA from the brain atlas. **c** Schematic

of optogenetic experiment. **d** Brain states (upper) and probability (bottom) before, during, and after laser stimulation of VLM glutamatergic terminals in the POA of awake animals ($n = 7$). Error bands, 95% bootstrap confidence intervals. **e** Bout durations (upper) and relative delta power (bottom) in optogenetic-induced and natural NREM sleep ($n = 7$ animals, **$P = 0.0003$ for bout duration, $P = 0.216$ for delta power, n.s. no significance, two-sided unpaired $t$-test). Error bars, SEM. Source data are provided as a Source Data file.

To avoid nonspecific activation, and more importantly to specifically activate the VLM-POA pathway, we performed terminal stimulation in Vglut2-Cre animals. Firstly, to characterize the projection patterns of VLM glutamatergic neurons in the POA, we performed anterograde tracing by unilaterally injecting AAV-FLEX-tdTomato into the VLM of Vglut2-Cre animals (Fig. 8a). The tdTomato-labeled neurons projected to several regions in the POA with particularly dense innervation into the VLPO (Fig. 8b). Then, we injected AAV-DIO-ChR2-eYFP into the VLM and implanted an optical fiber above the VLPO (Fig. 8c). We found that optogenetic activation of VLM glutamatergic terminals in the POA initiated NREM sleep in awake animals following the onset of light stimulation (Fig. 8d). The bout durations were

shorter in optogenetic-induced NREM sleep, compared to that in natural NREM sleep, whereas the delta power was not significantly different between two groups (Fig. 8e). In addition, we did not observe significant locomotor activity upon terminal stimulation in sleeping mice (Supplementary Fig. 12b). Notably, the wake–sleep transition probability in terminal stimulation was lower than that in soma stimulation (Fig. 8d and Fig. 7c), suggesting that other downstream targets of VLM glutamatergic neurons might be involved in NREM sleep control. Indeed, our anterograde tracing data demonstrated that in addition to the POA, VLM glutamatergic neurons also projected to other sleep-related brain regions, including the zona incerta (ZI)[37], the vlPAG, and the PIII (Supplementary Fig. 13). Optogenetic activation of

VLM glutamatergic neurons induced strong cFos expression in VLM, POA, and vlPAG (Supplementary Fig. 14). To further examine the VLM-POA pathway, we then performed pharmacological inhibition by infusion of the AMPA receptor antagonist NBQX[38,39] in the POA while optogenetically stimulating glutamatergic neurons in the VLM of Vglut2-Cre mice. We found that pharmacological inhibition of POA neurons suppressed optogenetic-induced wake-to-NREM transitions and reduced subsequent NREM sleep time (Supplementary Fig. 15). These data suggest that the POA is a major downstream target of VLM neurons and is required for VLM's function in NREM sleep control. In addition, our pharmacological data provides another piece of evidence to support glutamate transmission between VLM and POA in sleep regulation. Together, these results indicate that the VLM-POA circuitry plays an important role in the transitions from wakefulness to NREM sleep.

## Discussion

In this study, we identified a population of glutamatergic neurons in the ventrolateral medulla that are active during NREM sleep. Retrograde and anterograde tracing demonstrated that these neurons project to the POA, a prominent sleep-promoting center. Mcioendoscopic calcium imaging revealed that these VLM glutamatergic neurons displayed increased neuronal activity during the transitions from wakefulness to NREM sleep. Using chemogenetic manipulation, we demonstrated that these POA-projecting VLM neurons are both necessary and sufficient for the transitions from wakefulness to sleep. Furthermore, we showed that optogenetic activation of medulla glutamatergic neurons or their axonal terminals in the POA initiates NREM sleep in awake animals. Together, our results have uncovered an excitatory brainstem-hypothalamic circuit that controls the transitions from wakefulness to NREM sleep (i.e., sleep initiation). Notably, our finding of sleep-promoting excitatory glutamatergic neurons in the medulla differs from the known sleep-promoting GABAergic neurons previously reported in the POA[10–13], BF[17], PZ[15,16], and nNOS+ cortical neurons[19,20].

Previous studies have shown that the medulla is involved in regulating REM sleep and motor atonia[40,41]. For instance, loss of glutamate release (but not GABA/glycine) in the supraolivary medulla, a restricted region within the ventromedial medulla, resulted in exaggerated muscle twitches during REM sleep[37]. As indicated by our retrograde tracing experiments, the POA-projecting glutamatergic neurons are located more laterally (Fig. 3 and Supplementary Fig. 4), compared to the REM-related neurons in the ventromedial medulla. In microendoscopic calcium imaging experiments, we observed that a population of glutamatergic neurons are selectively active during REM sleep. Whether these VLM neurons can promote REM sleep or regulate muscle atonia remains to be tested. In both chemogenetic (Fig. 6) and optogenetic experiments (Fig. 7), we did not observe significant changes of REM sleep following the activation of VLM glutamatergic neurons. We believe that neural activity in other brain regions—possibly unrelated to the VLM—is needed to facilitate the transitions from NREM to REM sleep. In response to chemogenetic inhibition, we did observe a slightly decrease of REM sleep (Fig. 5e). However, we reason that this decrease is a consequence of reduced NREM sleep. Interestingly, REM sleep-controlling GABAergic neurons have also been identified in the ventral medulla[42], which is adjacent to the VLM glutamatergic neurons. In optogenetic experiments, we did observe that activation of VLM GABAergic neurons in some of the sleeping animals promoted REM sleep (Supplementary Fig. 9b), possibly due to the viral spread into the adjunct region. Intriguingly, the co-existence of REM sleep-promoting and NREM sleep-promoting neurons in the medulla raises the possibility of interactions among these neurons and highlights the importance of the medulla in sleep regulation.

Interestingly, NREM sleep initiated by optogenetic activation of VLM neurons usually lasted a few minutes and showed similar features to that of natural sleep (Fig. 7). This sleep phenomenon is distinct from previous optogenetic studies in the POA[18] and PIII[21], where the effect of optogenetic activation quickly disappears when laser stimulation is switched off. It suggests that sustained stimulation is required for the POA or PIII neurons to promote and more importantly to maintain NREM sleep, whereas sustained stimulation is not necessary to maintain NREM sleep in our case. Our data suggest the VLM glutamatergic neurons may function as an initiator to trigger the wake–sleep transition process, and subsequently their activity may not be required for NREM sleep maintenance. Compared to soma stimulation in the VLM, we noticed that terminal stimulation in the POA was less effective to initiate NREM sleep, considering both lower wake–sleep transition probability and shorter bout durations (Figs. 7, 8). In addition, our pharmacological experiments demonstrate that POA inhibition significantly suppressed but not fully eliminated wake-to-NREM transitions induced by VLM soma optogenetic stimulation (Supplementary Fig. 15). These results imply that VLM neurons may project to other brain areas to orchestrate the sleep initiation process. Indeed, our anterograde tracing and optogenetic stimulation experiments revealed several other downstream targets previously reported in regulating NREM sleep (Supplementary Fig. 13). It will be also important to identify the inputs of these VLM glutamatergic neurons. Particularly, effort should be put on neurons involved in wake/sleep regulation, to examine how NREM-promoting VLM neurons are activated during and prior to NREM sleep, and how they are inhibited during other brain brains. In addition to neuronal inputs, investigation of non-neuronal inputs, such as somnogens released from glial cells in the region might also be equally important to understand the sleep initiation process. Together, such functional studies of inputs and outputs of VLM neurons will yield mechanistic insight into sleep state transitions and maintenance.

## Methods

### Animals
All procedures were carried out in accordance with the US National Institute of Health (NIH) guidelines for the care and use of laboratory animals, and approved by the Animal Care and Use Committees of Columbia University. Both male and female adult mice (8–20 weeks old) were used for all experiments. The following mouse lines were used in the current study: C57BL/6J (JAX 000664), VGlut2-IRES-Cre (JAX 028863), Gad2-IRES-Cre (JAX 0101802), Ai9 (JAX 007909). Mice were housed in 12-hour light-dark cycles (lights on at 07:00 a.m. and off at 07:00 p.m., temperatures of 65–75 °F with 40–60% humidity) with free access to food and water.

### Viral constructs
AAV1-EF1α-double floxed-hChR2(H134R)-EYFP-WPRE-HGHpA, AAV1-Ef1α-DIO-EYFP, AAV1-Syn-FLEX-GCaMP6s, AAV1-Syn-FLEX-GCaMP6f, AAV8-hSyn-DIO-hM3D(Gq)-mCherry, AAV9-hSyn-DIO-hM4D(Gi)-mCherry, AAVrg-hSyn-Cre-WPRE-hGH, AAV9-CamKIIa-ChR2-eYFP, AAV1-CAG-FLEX-tdTomato, AAV1-mDlx-NLS-mRuby2 were obtained from Addgene. AAV1-EF1a-FLEX-TVA-mCherry from UNC vector core, AAV1-FLEX-2A-G(N2C)-mKate and RABV-ΔG -GFP-EnvA, a gift from Charles Zuker.

### Surgical procedures
Mice were anaesthetized with a mixture of ketamine and Xylazine (100 and 10 mg kg⁻¹, intraperitoneally), then placed on a stereotaxic frame with a closed-loop heating system to maintain body temperature. After asepsis, the skin was incised to expose the skull and a small craniotomy (~0.5 mm in diameter) was made on the skull above the regions of interest. A solution containing 50–200 nl viral construct was loaded into a pulled glass capillary and injected into the target region using a Nanoinjector (WPI). Optical fibers (0.2 mm diameter, 0.39 NA, Thorlabs) were implanted into the target region with the tip 0.4 mm above

the virus injection site for optogenetic manipulation. For EEG and EMG recordings, a reference screw was inserted into the skull on top of the cerebellum. EEG recordings were made from two screws on top of the cortex 1 mm from midline, 1.5 mm anterior to the bregma and 1.5 mm posterior to the bregma, respectively. Two EMG electrodes were bilaterally inserted into the neck musculature. EEG screws and EMG electrodes were connected to a PCB board which was soldered with a 5-position pin connector. All the implants were secured onto the skull with dental cement (Lang Dental Manufacturing). After surgery, the animals were returned to home-cage to recover for at least two weeks before any experiment.

For retrograde tracing, 150–200 nl AAVrg-hSyn.Cre.WPRE.hGH was unilaterally or bilaterally injected into the ventrolateral preoptic area (VLPO, Bregma 0.1 mm, lateral 0.9 mm, ventral 5.4 mm) of Ai9 mice. For rabies tracing, 200 nl mix of AAV-FLEX-G(N2C)-mKate and AAV-FLEX-TVA-mCherry (1:1) was unilaterally injected into the VLPO of Gad2-Cre mice. Two weeks after AAV injection, 200 nl RABV-ΔG-GFP-EnvA was unilaterally injected into the same VLPO. For anterograde tracing, 50 nl AAV1-CAG-FLEX-tdTomato was unilaterally injected into the ventrolateral medulla (VLM, Bregma −6.9 mm, lateral 1.1 mm, ventral 5.6 mm) of Vglut2-Cre mice. The ventral coordinates listed above are relative to the pial surface.

For chemogenetic inhibition, 200 nl AAV9-hSyn-DIO-hM4D(Gi)-mCherry was bilaterally injected in VLM, 200 nl AAVrg-hSyn.Cre.WPRE.hGH was bilaterally injected into the VLPO of C57BL/6J mice. For chemogenetic activation, 200 nl AAV8-hSyn-DIO-hM3D(Gq)-mCherry was unilaterally injected into the VLM, 200 nl AAVrg-hSyn.Cre.WPRE.hGH was bilaterally injected into the VLPO of C57BL/6J mice.

For optogenetic activation experiments, 200 nl AAV1-EF1α-double floxed-hChR2(H134R)-EYFP-WPRE-HGHpA was unilaterally injected into the VLM, an optical fiber implanted 0.4 mm on top of the viral injection site in Vglut2-cre mice. For terminal stimulation, 200 nl AAV1-EF1α-double floxed-hChR2(H134R)-EYFP-WPRE-HGHpA was unilaterally injected into the VLM of Vglut2-Cre mice, and an optical fiber was implanted in the POA. In in vivo pharmacological inhibition experiments, Vglut2-Cre mice were unilaterally injected with AAV1-DIO-ChR2-EYFP and implanted with an optical fiber in the VLM, then implanted with a double guide cannula (26 gauge, P1 Technologies) bilaterally above the POA. A metal head-post was attached during surgery for head fixation during the intracranial infusion.

For slice recording, 250 nl AAV1-mDlx-mRuby2 was unilaterally injected in the VLPO, 250 nl AAV1-DIO-ChR2-eYFP was contralaterally injected in the VLM of Vglut2-Cre mice.

For fiber photometry in GAD2-Cre mice, 200 nl AAV9-CaMKIIa-ChR2-eYFP was unilaterally injected in the VLM, 200 nl AAV1-FLEX-GCaMP6s was contralaterally injected in the VLPO. An optical fiber was implanted 0.2 mm above the VLPO injection site.

For microendoscopic calcium imaging, 200 nl AAV1-FLEX-GCaMP6f was unilaterally injected in the VLM of Vglut2-Cre mice. A GRIN lens (0.5 or 0.6 mm diameter, Inscopix) was implanted 0.2 mm above the injection site. After >3 weeks, the cover was removed to expose the GRIN lens and a miniaturized, single-photon, fluorescence microscope (Inscopix) was lowered over the implanted GRIN lens until the GCaMP6f fluorescence was visible under illumination with the microscope's LED. The microscope's baseplate was then secured to the skull with dental cement darkened with carbon powder for subsequent attachment of the microscope to the head. After recovery from surgery, we did not observe any gross behavioral abnormality, and these mice exhibited normal sleep–wake cycles.

## Sleep recording

Mouse sleep behavior was monitored using EEG and EMG recording along with an infrared video camera at 30 frames per second.

Recordings were performed for 24–48 h (light on at 7:00 a.m. and off at 7:00 p.m.) in a behavioral chamber inside a sound-attenuating cubicle (Med Associated Inc.). Animals were habituated in the chamber for at least 4 h before recording. EEG and EMG signals were recorded, bandpass filtered at 0.5–500 Hz, and digitized at 1017 Hz with 32-channel amplifiers (TDT, PZ5 and RZ5D or Neuralynx Digital Lynx 4S). Spectral analysis was carried out using fast Fourier transform (FFT) over a 5 s sliding window, sequentially shifted by 2 s increments (bins). Brain states were semi-automatically classified into wake, NREM sleep, and REM sleep states using a custom-written Matlab (version 2021, MathWorks) program[43] (wake: desynchronized EEG and high EMG activity; NREM: synchronized EEG with high-amplitude, delta frequency (0.5–4 Hz) activity and low EMG activity; REM: high power at theta frequencies (6–9 Hz) and low EMG activity). Semi-auto classification was validated manually by trained experimenters. Relative delta power was calculated by dividing the delta power in the 2-s bins by the total EEG power averaged across the recording session.

## Fiber photometry

Fiber photometry recordings were performed essentially as previously described[44]. In brief, Ca²⁺-dependent GCaMP fluorescence were excited by sinusoidal modulated LED light (465 nm, 220 Hz; 405 nm, 350 Hz, Doric lenses) and detected by a femtowatt silicon photo-receiver (New Port, 2151). Photometric signals and EEG/EMG signals were simultaneously acquired by a real-time processor (RZ5D, TDT) and synchronized with behavioral video recording. A motorized commutator (ACO32, TDT) was used to route electric wires and optical fiber. The collected data were analyzed by custom MATLAB scripts. They were first extracted and subject to a low-pass filter at 2 Hz. A least-squares linear fit was then applied to produce a fitted 405 nm signal. The DF/F was calculated as: (F-F0)/F0, where F0 was the fitted 405 nm signals. Data were smoothed using a moving average method and downsampled to 5 Hz. In Supplementary Fig. 6, photometric data were further normalized across animals using Z-score calculation.

## Optogenetic manipulation

All optogenetic stimulation were conducted unilaterally. Mice were habituated in the behavioral chamber for at least 4 hours before the experiment. Light pulses (20 Hz, 10 ms) with different durations (30 s, 1 min, 2 min) from a 473 nm laser diode (Shanghai laser & Optics Century Co., Ltd) were controlled by a microcontroller board (Arduino Mega 2560, Arduino). Laser power is set to 4–6 mW for somatic stimulation in the VLM and 10–15 mW for terminal stimulation in the VLPO. We used two methods to trigger laser stimulation: (1) wake-trigger stimulation: An IR-camera (30 fps) was placed on the ceiling to videotape animal behavior. A custom Matlab program[45] was used to real-time process video frames to detect animal's location by subtracting each frame from the pre-acquired background image (without the mouse). Laser stimulation was automatically triggered when animal movement continued for a period (5 min). A minimal interval of 30 min was set between trials. In no-light control experiments, the same trigger method was applied except the laser power was off. (2) fixed interval stimulation: inter-stimulation interval for optogenetic stimulation is fixed to 45 min. To compare optogenetic-induced and natural NREM sleep (Fig. 7e), we used the first NREM sleep episode (if existed) following laser stimulation in 15-min time windows (5 min before laser, 10 min after laser) as optogenetic-induced sleep, used all NREM sleep episodes from 15-min time windows between laser stimulation (at least 15 min away from previous or next stimulation) in the same recording sessions as natural sleep, and calculated bout durations and relative delta power of NREM sleep episodes.

## Microendoscopic calcium imaging

Imaging sessions took place during the light cycle in a behavioral chamber placed within a sound-attenuated cubicle (Med. Associates).

Calcium activity was acquired using the nVista 3.0 hardware and IDPS software (Inscopix) with 475 nm LED illumination (10 Hz, 0.4–1.2 mW/mm²). EEG and EMG were acquired using Neuralynx Digital Lynx 4S controlled by a custom-built MATLAB program via Neuralynx API. A TTL signal delivered from the Inscopix system to the Neuralynx system throughout the recording session was used to synchronize the timing between the imaging and EEG/EMG recordings. Each recording session lasted 60–120 min, and for each mouse the data from a single recording session was included. Imaging data were processed in IDPS (Inscopix, version 1.6.0.3225) and MATLAB[46]. First, the acquired images were spatially down-sampled by a factor of 4. To correct for lateral motion of the brain relative to the GRIN lens, we used the motion correction function in IDPS, as in previous studies[47,48]. Regions of interest (ROIs) were then manually identified. The pixel intensities within each ROI were averaged to create a fluorescence time series. For individual neurons, the DF/F was calculated as the difference between the calcium activity at each bin and the averaged calcium activity of the whole recording time, divided by the average. To quantify the calcium activity, we used the OASIS fast deconvolution algorithm[49]. The identified events were reviewed along with calcium imaging video by an experimenter and motion-induced artifacts were excluded for further analysis. To compare the activity in different brains states, we used the integrated area under curve (AUC) of detected events and normalized it to the durations (in minute) of each state, which yields relative activity per minute. The N-W or R-W selectivity index was calculated as:

$$\text{Index} = (\text{AUC}_a - \text{AUC}_b)/(\text{AUC}_a + \text{AUC}_b)$$

where $\text{AUC}_a$ and $\text{AUC}_b$ refer to AUC activity in brain state a (e.g., NERM sleep) and b (e.g., wake) respectively. Index ranges from −1 to 1, with 0 indicating no selectivity between two states.

For the analysis of calcium activity during the transitions (Fig. 2d), we calculated AUC activity in the NREM active cells before and after wake-to-NREM or NREM-to-wake transitions. A 30-s window in each brain state (e.g. 30-s wake, 30-s NREM for W-N transitions) was used for quantification. Transitions with less than 30s-episodes in either state were excluded for analysis.

## Chemogenetic manipulation

After habituation for 12 h in the testing chamber, C57BL/6J mice expressing hM4Di or hM3Dq in the VLM were injected with saline (day 1) and CNO (day 2, 1 mg/kg body weight) intraperitoneally (i.p.) at the same time of the days. Injections were performed in light cycles (10:00 a.m.) for chemogenetic inhibition and in dark cycles (9:30–10:00 PM) for chemogenetic activation. In control experiments, wild type mice without viral injection were treated with CNO and saline either in light cycles or dark cycles. Sleep recording started at least 1 h before saline injection and lasted 24 h after CNO injection. EEG and EMG in the time window (0–12 h after CNO or saline injection) were used for data analysis.

## Pharmacological inhibition

On the day of the experiment, mice were habituated and tested with photostimulation in a behavioral chamber for overnight (pre-test). Following the pre-test, the selective AMPA receptor antagonist NBQX (5 mg/ml in 0.9% NaCl, 0.25 µl, Tocris Bioscience) was bilaterally infused into the POA using two 1 µl microsyringes (Hamilton) and two internal cannulas (P1 Technologies) inserted into the double guide cannula that implanted above the POA. The infusion rate was approximately 0.04 µl/min controlled by a syringe pump (Harvard Apparatus). As a control, saline (0.9% NaCl) was similarly applied. After the intracranial infusion, mice were moved back to the chamber for further optogenetic experiments. The first laser stimulation started 30 min after drug infusion. A total of 6–8 trials in the first 4 h was used for data analysis.

## Whole brain clearing and imaging

To identify neurons projecting to the POA, we unilaterally injected ~200 nl AAVrg-hSyn.Cre.WPRE.hGH in the VLPO of Ai9 reporter mice. Six weeks after viral injection, mice were perfused with phospho-buffered saline (PBS) containing 10 U/ml heparin, followed by 4% paraformaldehyde. Brains were then harvested and post-fixed in 4% paraformaldehyde for 3 h at room temperature. The whole brains were cleared by the CUBIC method as previously described[39,50]. Briefly, mouse brains were washed 3 times in PBS before immersion in CUBIC reagent-1 (diluted 1:2 in water) overnight, incubated in reagent-1 for 7–10 days. Then, brains were washed with PBS, degassed in PBS overnight, and immersed in reagent-2 (diluted 1:2 in PBS) for 6–24 h before incubated in reagent-2 containing TO-PRO-3 (1:5000, Thermo Fisher Scientific) for additional 7–10 days. Reagent 1 contained 25 wt% urea (Sigma- Aldrich), 25 wt% N,N,N',N'-tetrakis (2-hydroxypropyl) ethylenediamine (Sigma- Aldrich) and 15 wt% Triton X-100 (Nacalai Tesque). Reagent 2 contained 50 wt% sucrose (Sigma-Aldrich), 25 wt% urea, 10 wt% triethanolamine (Sigma-Aldrich) and 0.1% (v/v) Triton X-100. All clearing procedures were performed at room temperature with gentle shake to prevent sample deformation caused by temperature fluctuation and fluorescence loss. Reagent 1 and Reagent 2 were refreshed every 3 days. Samples were imaged in an oil mix (mineral oil and silicone oil 1:1) horizontally from ventral to dorsal by Light-sheet fluorescence microscopy (UltraMicroscope, LaVision Bio-Tec) as previously described[39]. The images were acquired with a z-step size of 5 µm. Exposure time was 50 ms per channel per z step. Data was processed in ImageJ (version 1.52i). The gamma value of the images was set to 0.5 for display purposes. The whole-brain data was registered to a reference atlas (Allen Brain Institute, 25-µm resolution volumetric data with annotation map, http://www.brain-map.org) using elastix (version 5.0)[51,52]. The voxel size of both sample data and reference template were scaled to 6.5 µm. The 3D reconstruction, cell tracing, and structure labeling were performed in Imaris (version 9.6, Bitplane).

## Fos induction and immunohistochemistry

In sleep deprivation experiments, mice were allocated into three groups: (1) control, (2) sleep deprivation (SD), (3) recovery sleep (RS) after deprivation. Each group was placed in a behavioral chamber, which was further contained in a sound-attenuating cubicle (Med Associates Inc). Mice were habituated for 48 h before Fos induction. For sleep deprivation, a stepper motor controlled by an Arduino board was used to move a rod to sweep the floor from one side to the other at a speed of 2 cm/s. As a motor control, Mice in the first group were subject to sweeping movement for 2 h before perfusion. SD mice were subject to 30-s sweeping movement followed by a 90-s interval for 6 h, and then sacrificed for perfusion. RS mice were perfused 2 h after 6-h SD. All sleep manipulation was carried out in the light cycle, and animals were sacrificed between 4:00 p.m. and 6:00 p.m. Mouse brains were sectioned coronally at 100 µm and processed for immunostaining as previously described[53]. Tissue sections were incubated with guinea pig c-Fos antibody (1:1000, Synaptic Systems, cat#226308) for 24 h at 4 °C. Fluorescently tagged secondary antibodies (Alexa-488 donkey anti-guinea pig, 1:500, Jackson ImmunoResearch, cat#706-545-148) were used to visualize Fos expression. All sections were imaged using a Zeiss 810 confocal microscope. Cell counting (brain sections from AP-6.5 mm to AP-7.2 mm) was performed manually in ImageJ.

In optogenetic stimulation experiments (Supplementary Fig. 14), Vglut2-Cre mice were injected with AAV-DIO-ChR2-eYFP unilaterally in the VLM. To maximize the c-Fos signals, a 30-min laser stimulation (20 Hz, 4–5 mW) was applied. Mice were kept in the recording chamber for additional 40 minutes after stimulation, then perfused for c-Fos

staining as described above. The injection site and other sleep/wake related brain regions were examined for c-Fos expression.

## Fluorescence in situ hybridization (FISH)

Mice were deeply anaesthetized, and fresh frozen brains were sectioned at 20 μm thickness using a cryostat. FISH was performed using RNAscope Multiplex Fluorescent Assay V2 (Advanced Cell Diagnostics). Reagents: *Fos* in situ hybridization probe: cat# 316921, *Slc17a6* in situ hybridization probe: cat# 319171, *Slc32a1* in situ hybridization probe: cat# 319191. Tyrosine hydroxylase (TH) in situ hybridization probe: cat# 317621 (Advanced Cell Diagnostics). Images were acquired using a Zeiss 810 confocal microscope. Cell counting (brain sections from AP-6.5 mm to AP-7.2 mm) was performed manually in ImageJ.

## Histology

Viral expression and placement of optical implants were verified at the termination of the experiments using DAPI counterstaining of 100 μm coronal sections (Prolong Gold Antifade Mountant with DAPI, Invitrogen). Images were acquired using a Zeiss 810 confocal microscope. Cell numbers were counted manually in ImageJ.

## Slice recording

Four weeks after viral injection, mice were quickly decapitated under isoflurane sedation. Brains were placed in artificial cerebral spinal fluid (aCSF). Coronal slices (300 μm) containing the POA were cut on a vibratome (Leica VT1200) in sucrose cutting solution containing 2.5 mM KCl, 10 mM $MgCl_2$, 0.5 mM $CaCl_2$, 1.25 mM $NaH_2PO_4$, 26 mM $NaHCO_3$, 234 mM sucrose, 11 mM glucose. Slices were then transferred to aCSF containing 126 mM NaCl, 26 mM $NaHCO_3$, 2.5 mM KCl, 1.25 mM $NaH_2PO_4$, 2 mM $CaCl_2$, 1 mM $MgCl_2$, and 10 mM D-glucose bubbled with 95% $O_2$ and 5% $CO_2$. Slices were incubated at 34 ± 1 °C for 30 min before resting at room temperature prior to the experiment. Regular pipette intracellular solution contained 127 mM potassium gluconate, 8 mM NaCl, 4 mM ATP-Mg, 0.6 mM EGTA, 0.3 mM GTP, 10 HEPES, and 8.1 mM biocytin adjusted to pH 7.3–7.4 with KOH. Osmolarity was adjusted to 290–300 mOsm.

Inhibitory GABAergic cells in the POA were located using the presence of mRuby fluorescence. Cells were targeted approximately 20–100 μm from the slice surface. Patch-clamp Scientifica MicroStar micromanipulators were controlled using LinLab2 software (Scientifica). Recordings were performed in juxtacellular or whole-cell mode with MultiClamp 700B amplifiers (Molecular Devices) in the current clamp or voltage clamp mode at 34 ± 1 °C bath temperature. Data acquisition was performed through an Axon Digidata 1550B (Molecular Devices) connected to a PC running pClamp 11 (Molecular Devices). Recordings were sampled at 10 kHz and filtered with a 2-kHz Bessel filter. Patch pipettes were pulled with a Flaming/Brown micropipette puller P-80PC (Sutter Instruments) and had an initial resistance of 3–8 MΩ. Series resistance was automatically compensated at 15 MΩ. The membrane potential values given were not corrected for the liquid junction potential which was approximately −16 mV. Cells were stimulated with blue LED light through a 40× water-immersion objective using pE-300ultra (CoolLED Limited) at approximately -18.8 mW. Stimulation protocols comprised of 10 sweeps of paired pulses at 2, 5, 10, 20, and 40, and 60 Hz delivered with a light on time of 1 ms. 6-cyano-7-nitro-quinoxaline-2,3-dione (CNQX; 10 μM) and DL-2-Amino-5-phosphonopentanoic acid (DL-AP5; 10 μM) was bath applied in some recordings.

Unitary excitatory post-synaptic potentials (EPSPs) and post-synaptic currents (EPSCs) were analyzed using custom Matlab scripts. EPSPs or EPSCs were considered if the absolute value is >2 times of the standard deviation of the baseline. Baseline was calculated as the mean of 10 ms prior to the stimulation or pre-synaptic action potential. Amplitudes were found at the maximum or minimum point within a 1.5–18 ms time window after the stimulation onset. Lower- or

upper-bounds of detection windows were adjusted when necessary. The rise time was determined as the time interval encompassing 20–80% of the amplitude. Latencies were determined by calculating the onset time of the PSP or PSC and subtracting the stimulation onset. This onset time represented the intersection of the line at baseline and the line through the 20% and 80% amplitude points.

## Statistics

No statistical methods were used to predetermine sample size, and investigators were not blinded to group allocation. No method of randomization was used to determine how animals were allocated to experimental groups. Mice in which post hoc histological examination showed viral targeting or fiber implantation was in the wrong location were excluded from analysis. Two-sided paired *t*-test, unpaired *t*-test, and Mann–Whitney *U*-test were used and indicated in the respective Figure legends. Values are tested against normal distributions using the Kolmogorov–Smirnov test with statistical significance set at $p < 0.05$. All analyses were performed in MATLAB. Data are presented as mean ± SEM.

## Reporting summary

Further information on research design is available in the Nature Research Reporting Summary linked to this article.

## Data availability

All other raw data supporting the findings of this study are available from the corresponding author upon request. Data will be delivered within 2–4 weeks once requested. Source data are provided with this paper.

## Code availability

Custom scripts for calcium imaging (https://doi.org/10.5281/zenodo.6870710), EEG/EMG (https://doi.org/10.5281/zenodo.6870666), and behavioral analysis (https://doi.org/10.5281/zenodo.6870689) have been deposited in Zenodo and cited in the references.

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

## Acknowledgements

We thank Charles Zuker at Columbia University for his support in the early stage of the study, for his help with microendoscopic imaging experiments. We thank Xiaoke Chen at Stanford University for the gift of the retrograde virus. We acknowledge the Cellular Imaging Core at the Columbia Zukerman Institute for access to their facility. We also thank Mingyu Ye, Xiaoke Chen, Gergely Turi, and Charles Zuker for helpful discussions. This work was supported by startup funds from Columbia University to Y.P., Columbia University Precision Medicine Initiative to Y.P. and C.D.M., and by the Swiss National Science Foundation Postdoctoral Fellowship to J.S.

## Author contributions

S.T. and Y.P. designed the study, carried out the experiments, and analyzed data, F.Z. performed histology and circuit tracing experiments. L.W. performed imaging experiments. J.C.S. performed behavioral experiments. J.S. and C.M. performed slice recording experiments and

analysis. X.C. performed data analysis. H.J. performed rabies-based tracing experiments. Y.P. and S.T. wrote the paper.

## Competing interests

The authors declare no competing interests.
