## [Peer Review File · Nature Communications]

Control of Non-REM sleep by ventrolateral medulla glutamatergic neurons projecting to the preoptic areaReviewers' comments:

Reviewer #1 (Remarks to the Author):

In this original article, Teng and colleagues report that a restricted population of glutamatergic neurons in the ventral lateral medulla (VLM) is involved in wake-NREM sleep transitions. They described that these neurons expressed cFos after frequent wake-sleep transitions induced by a novel sleep-deprivation method and that the activation of RVM glutamate neurons or a subpopulation projecting to the POA promotes sleep. The authors also tested their causal role in sleep induction using loss-of-function approaches. It has been long speculated that sleep-promoting neurons are also present in the medulla. Parafacial zone was recognized a few years ago and Teng and colleagues now report another medullary sleep-promoting cell group. These results are quite interesting and will significantly advance our understanding on the sleep regulatory mechanisms. However, the methods used (chemo vs opto for different experiments) and results are inconsistent across experiments. Also, an increase in wake-to-sleep transitions in chemoactivation experiments would also mean an increase sleep-to-wake transitions. Increases in sleep amounts are not observed soon after CNO injections, and the latencies to sleep after activation or inhibition of VLM neurons (all experiments) are not reported. Collectively, the presented data leaves one wondering what the precise role of VLM glutamate neurons and the VLM-POA circuit in sleep-wake behavior could be rather than providing a clear answer. Also, the specificity and selectivity of the AAVs were not described in detail, which is crucial (quantification is missing). A major revision may significantly improve the quality of the findings.

1. The novel method of sleep deprivation used in the study is interesting, but cFos expression from these experiments cannot distinguish between sleep-wake and wake-sleep transitions as there were increases in both types of transitions. While it is true that activation of glutamatergic neurons promotes sleep, not wake, higher cFos was not observed after sleep rebound (as one would expect). Moreover, photoactivation of these neurons induced sleep did not induce sleep during stimulations but only after the end of stimulations. Hence, it is difficult to describe the specific contribution of these neurons in sleep-wake behavior or sleep-wake transitions in particular. Understanding in vivo activity pattern of VLM glutamatergic neurons across sleep-wake (electrophysiology/ calcium imaging) will be helpful in this regard. This information will substantially improve the significance of the Fig 1 data and the manuscript as a whole.
2. There is no doubt that the POA is a crucial sleep-promoting center, but there are regions within the brainstem, for example, the parafacial zone can promote NREM sleep. Also, classical transection studies indicate that the brainstem and forebrain can independently control sleep-wake states. Did the authors test the possibility that these medullary glutamatergic neurons activated the PZ to promote sleep? If not, how did they rule out this possibility?
3. Did the anterograde tracing from medullary glutamatergic neurons show any terminals in the PZ or in other sleep-promoting regions viz, VTA (GABA neurons), Nucleus accumbens or midbrain tegmental region, VLPAG etc?
4. Line 103-106 – Did the authors repeat the sleep deprivation experiments in these mice? It is not very clear from the description.
5. AAV-CaMKIIa-ChR2 – specificity/selectivity is not clear; In my experience, transfection of CaMKinase AAVs show differential expression in the cortical vs subcortical structures (quite variable). A histology figure of the medullary injection site and the quantification of their selectivity is crucial to better appreciate the findings.
6. Fig 3 C, D – it appears that 2 animals producing strong effects and others had minor effects on wake or NREM sleep. Quantification of transduced neurons/area in both medulla and the POA would be helpful to understand such variation. The number and duration of wake bouts were not presented. I also request the authors to verify the significance levels for the first 2-h bin of wake and NREM in Fig 3C.
7. Fig 4: CNO effects are expected to begin with 10-15 mins after the injections; but, CNO-activation had no effects on sleep in the first 2 hrs. If these neurons were to promote wake-sleep transitions, one would also expect a decrease in sleep latency after their activation, which is not probably observed. No latency data is presented.
8. Fig 5 and supplementary fig 6: I am wondering if the high frequency stimulations (20Hz) have caused 'depolarization blockade' in the neurons which resulted in increased sleep after the stimulations were stopped. Did the authors try to apply low frequency stimulations or chemoactivations?

Reviewer #2 (Remarks to the Author):

The neural circuits and action mechanisms through which the brain promotes and maintains sleep and wakefulness have not been fully resolved yet. The paper by Teng et al. investigates the role ventrolateral medulla (VLM) glutamatergic neurons projecting to the preoptic area (POA) in the control of wake/sleep transitions by using cutting-edge genetically engineered systems. They found that chemogenetic inhibition of POA-projecting medulla neurons increases wakefulness, whereas activation produces NREM sleep. Moreover, optogenetic activation of medulla glutamatergic neurons or their projections in the POA initiates NREM sleep in awake mice. These are potentially important findings; however, I am not impressed yet because the observed phenotypes are inconsistent between the experiments and in-vivo activity of VLM neurons across sleep/wake states is unknown. More specifically, chemogenetic inhibition of retrogradely labeled VLM neurons increases wakefulness and suppresses NREM and REM sleep (total time and bouts), whereas activation only induces NREM sleep (time and bouts) and has no effect on REM sleep. Furthermore, optogenetic activation of VLM glutamatergic neurons increases NREM sleep after the stimulation in awake mice, whereas activation of VLM glutamatergic terminals in the POA apparently induces NREM sleep during stimulation (or has no effect in many trials). On the other hand, activation of VLM glutamatergic neurons or POA terminals has a wake-inducing or no effect, respectively, in sleeping mice. There is quite a bit of speculation in the paper to explain the inconsistencies. Thus, I feel that the authors need more time to sort this out before coming up with a solid conclusion for the role of POA-projecting VLM glutamatergic neurons in sleep/wake regulation.

I have the following specific comments/suggestions:

- Using retrograde labeling, the authors found that around 59.4% are glutamatergic neurons, and around 0.9% are GABAergic neurons projecting to the POA. Still there are around 40.6% neurons projecting from the VLM to POA. What type of neurons are the latter ones? How do they contribute to the wake-sleep transitions?
- I think it is important to measure the activity of the VLM neurons (especially the ones projecting to the POA) to see what sleep/wake behavior they seem to track. Without that, it is just another manipulation that may or may not inform our understanding of the animal's natural behavioral repertoire. This may also clarify the inconsistencies for the timing of NREM sleep induction in the optogenetic experiments. Judging from supplementary figure 3, the authors have the ability to conduct in-vivo Ca imaging with fiber photometry.
- For chemogenetic experiments (figures 3 and 4), the authors used retrograde labeling which will label both glutamatergic (59.4%) and non-glutamatergic (40.6%) neurons, but the authors used vGlut2-Cre mice which will only label glutamatergic neurons (not all of these neurons projecting to the POA) for optogenetic manipulation (figures 5 and 6). I think it would be meaningful if the authors also use vGlut2-Cre mice for the chemogenetic experiments.
- In figure 3, the chemogenetic inhibition experiment has two strong outliers, which surely enhance the mean values. They should provide a meaningful explanation why these two animals are so different from the other 5 animals in the group. The authors should also show the profiles of hourly sleep/wake amounts during the recording time for the chemogenetic experiments shown in figures 3 and 4.
- The dose of 1 mg/kg used for all chemogenetic experiments appears to be arbitrary without explanation. Since this is a pharmacologic experiment, the responses to different CNO doses should be evaluated.
- The authors speculate in lines 215-221 that viral spread and non-specific optogenetic stimulation of neurons in the lateral paraventricular nucleus induces locomotor activity suppressing sleep during stimulation. I think this hypothesis can easily be tested by retrogradely labeling VLM

neurons with channelrhodopsin, for example, from the VLPO to avoid labeling of glutamatergic neurons in the lateral paragigantocellular nucleus.

- Do the glutamatergic neurons regulate wake-sleep transition due to glutamate transmission or do these neurons co-release other transmitters?

Minor comments:

- In figure 1d the Slc32a1 staining is hard to see.
- Figure 3b, correct 10:00ar to 10:00am.

Reviewer #3 (Remarks to the Author):

In their manuscript, the authors first combined sleep deprivation using a rod moving in the cage for 30s every 90s during 6 hours. Then they show that more cFos+ cells are localized in a region lateral to the LPGi during sleep deprivation compared to control and sleep recovery. From the Figure, a fragmentation of sleep is visible but I don't see the quantification of each states (NREM, REM and W) quantities. Without such information it is not possible to prove that there is a sleep deprivation and recovery. It is required that the authors provide such analysis with statistical analysis of the quantities. Statistical analysis of the number of cFos+ neurons is also necessary. The labeling of slc32a1 is quite surprising since it does not show the presence of neurons and it is well known that this area contain a large number of GABAergic neurons. In addition, the area they are showing is corresponding to the LPGi not a region lateral to it (line 84).

Line 101, A1 and C1 as part of the LPGi not outside as stated.

In summary, it is required that the authors make a complete sleep deprivation and recovery and demonstrate that activation of the cells is increased during NREM and not during sleep deprivation. Indeed, activation during sleep deprivation would rather mean the cells are Wake-active.

The authors then use advanced specific tract tracing methodology to demonstrate that glutamatergic neurons of the LPGi excite GABAergic neurons of the POA. Overall, the results seem quite convincing but I'm lacking number of mice used in the text.

Figure 2b is not clearly showing where are located the neurons. A delimitation of the structures is needed there. The same in C, what bregma levels it correspond to?

Figure 3e, neurons retrogradely labeled are also not well localized.

Line 130, the fiber photometry experiment is not fully convincing since stimulation of glutamatergic neurons of the LPGi might induce NREM by another projection than the one to the POA. In such case, GABAergic neurons in the POA would also be activated.

In the following part of the paper, the authors inactivated using DREADDS the POA-projecting LPGi neurons and studied its effect on sleep. They obtained a small decrease of NREM and REM and an increase in W. They concluded that « POA-projecting VLM neurons are required for the transitions from wakefulness to NREM sleep ». They cannot write such sentence since there are still many transitions. Further, it is rather an increase in NREM than an effect on transitions.

They display Fig. 3 cumulative time, it is not adapted and they should show hour by hour values. Although an effect is seen it does not seem very strong. What is the percentage of decrease?

I have the same requests and concerns for the stimulation experiments. Then, the authors stimulate the glutamatergic neurons of the LPGi using optogenetic. Surprisingly, in this case they obtained a waking state. Moreover, when they stimulated during NREM they obtained wake with locomotion. They explained such results by the recruitments of glutamatergic neurons involved in waking. Unfortunately, these results are against their hypothesis of a role of these neurons in inducing NREM. To overcome the problem, they stimulate the glutamate terminals in the POA. In this case, they obtained a small increase in NREM. However, each trial was not followed by an induction of NREM. The results are therefore not very convincing.

Overall, my feeling is that some GLU neurons in the LPGi neurons might be implicated in promoting NREM but more convincing and specific data is needed to reach that conclusion.

Reviewer #1 (Remarks to the Author):

Specific comments:

1. *The novel method of sleep deprivation used in the study is interesting, but cFos expression from these experiments cannot distinguish between sleep-wake and wake-sleep transitions as there were increases in both types of transitions. While it is true that activation of glutamatergic neurons promotes sleep, not wake, higher cFos was not observed after sleep rebound (as one would expect). Moreover, photoactivation of these neurons induced sleep did not induce sleep during stimulations but only after the end of stimulations. Hence, it is difficult to describe the specific contribution of these neurons in sleep-wake behavior or sleep-wake transitions in particular. Understanding in vivo activity pattern of VLM glutamatergic neurons across sleep-wake (electrophysiology/ calcium imaging) will be helpful in this regard. This information will substantially improve the significance of the Fig 1 data and the manuscript as a whole.*

We thank the reviewer for the comments. We have now performed microendoscopic calcium imaging (Fig. 2) to examine in vivo activity of VLM glutamatergic neurons across sleep-wake. Indeed, our data shows that a population of VLM cells are selectively active during NREM sleep, compared to that in wake and REM sleep. In addition, we also show the quantification of cFos expression (Fig. 1c).

2. *There is no doubt that the POA is a crucial sleep-promoting center, but there are regions within the brainstem, for example, the parafacial zone can promote NREM sleep. Also, classical transection studies indicate that the brainstem and forebrain can independently control sleep-wake states. Did the authors test the possibility that these medullary glutamatergic neurons activated the PZ to promote sleep? If not, how did they rule out this possibility?*

As discussed in the manuscript, VLM glutamatergic neurons might project to other brain regions beside the POA. Indeed, our anterograde tracing revealed several downstream targets (Supplementary fig 13). Among them, we did observe some terminals in the PZ. However, using retrograde tracing from the PZ, we didn't observe clear labeled cells in the VLM, suggesting that cells in other regions (possibly, dorsal part of the medulla) project to the PZ.

3. *Did the anterograde tracing from medullary glutamatergic neurons show any terminals in the PZ or in other sleep-promoting regions viz, VTA (GABA neurons), Nucleus accumbens or midbrain tegmental region, VLPAg etc?*

Anterograde tracing showed terminals in several other sleep-promoting brain areas, including the ZI, PIII, and vIPAG (Supplementary fig 13), but not the VTA and Nucleus accumbens.

4. *Line 103-106 – Did the authors repeat the sleep deprivation experiments in these mice? It is not very clear from the description.*

Sorry for the lack of clarity. Yes, the same sleep deprivation was used in co-staining experiments.

5. AAV-CaMKIIa-ChR2 – specificity/selectivity is not clear; In my experience, transfection of CaMKinase AAVs show differential expression in the cortical vs subcortical structures (quite variable). A histology figure of the medullary injection site and the quantification of their selectivity is crucial to better appreciate the findings.

Thank you for the suggestion. We have performed co-staining experiments showing that AAV-CaMKII indeed targets predominately glutamatergic neurons in the VLM (~80%, Supplementary Fig. 6a). More importantly, we also showed that optogenetic activation of CaMKII-labeled cells in the VLM induced wake-sleep transitions at a similar level to that observed in Vglut2-Cre mice (Fig. 6f and Supplementary Fig. 10).

6. Fig 3 C, D – it appears that 2 animals producing strong effects and others had minor effects on wake or NREM sleep. Quantification of transduced neurons/area in both medulla and the POA would be helpful to understand such variation. The number and duration of wake bouts were not presented. I also request the authors to verify the significance levels for the first 2-h bin of wake and NREM in Fig 3C.

As requested, we have added quantification of transduced cells in Supplementary fig. 8a. The number and duration of wake bouts have been added to the revised Fig 4. We also calculated the first 2-h wake and NREM and verified the significance between CNO and saline (Supplementary fig. 8b).

7. Fig 4: CNO effects are expected to begin with 10-15 mins after the injections; but, CNO-activation had no effects on sleep in the first 2 hrs. If these neurons were to promote wake-sleep transitions, one would also expect a decrease in sleep latency after their activation, which is not probably observed. No latency data is presented.

We observed the effect of CNO on NREM sleep in the 2nd and 3rd hour (Fig. 5) after chemogenetic activation. As suggested, we have calculated latency, a trend but no statistical significance was observed in chemogenetic inhibition experiments (Supplementary fig. 8c). Similarly, no significance for latency in chemogenetic activation.

8. Fig 5 and supplementary fig 6: I am wondering if the high frequency stimulations (20Hz) have caused 'depolarization blockade' in the neurons which resulted in increased sleep after the stimulations were stopped. Did the authors try to apply low frequency stimulations or chemoactivations?

We have added data in Supplementary fig. 11d showing low frequency stimulation.

Reviewer #2 (Remarks to the Author):

Specific comments/suggestions:

- Using retrograde labeling, the authors found that around 59.4% are glutamatergic neurons, and around 0.9% are GABAergic neurons projecting to the POA. Still there are around 40.6% neurons projecting from the VLM to POA. What type of neurons are the latter ones? How do they contribute to the wake-sleep transitions?

Our previous quantification was based on viral expression. There are some potential limitations regarding infection efficiency. Also, some serotype of virus displayed bias to certain cell types. To better characterize the identity, we performed FISH in Ai9 mice injected with AAVrg-Cre. Our new data (Supplementary fig. 5) show that around 75% are glutamatergic, and again around 0.2% are GABAergic. Using FISH, we also tried to figure out the remaining 25% with different neuropeptide markers (e.g. CRH, CCK, and Carpt), suggested by their expression in the

medulla from Allen Brain database. So far, we haven't seen significant overlap between retrograde labeling and peptide staining.

- I think it is important to measure the activity of the VLM neurons (especially the ones projecting to the POA) to see what sleep/wake behavior they seem to track. Without that, it is just another manipulation that may or may not inform our understanding of the animal's natural behavioral repertoire. This may also clarify the inconsistencies for the timing of NREM sleep induction in the optogenetic experiments. Judging from supplementary figure 3, the authors have the ability to conduct in-vivo Ca imaging with fiber photometry.

Thank you for the comment. We previously did use fiber photometry to examine bulk activity of VLM glutamatergic neurons, and found mixed results (i.e. small but reliable activity during NREM sleep, increased activity during wake particularly when animals move), suggesting the heterogeneity of VLM neurons. Now, we have performed microendoscopic calcium imaging to measure the activity at single-cell resolution (Fig. 2 and Supplementary Fig. 3).

- For chemogenetic experiments (figures 3 and 4), the authors used retrograde labeling which will label both glutamatergic (59.4%) and non-glutamatergic (40.6%) neurons, but the authors used vGlut2-Cre mice which will only label glutamatergic neurons (not all of these neurons projecting to the POA) for optogenetic manipulation (figures 5 and 6). I think it would be meaningful if the authors also use vGlut2-Cre mice for the chemogenetic experiments.

We thank the reviewer for the suggestion. We tried chemogenetic activation experiments in a few Vglut2-cre mice, and observed increased locomotor activity after CNO treatments. As demonstrated by our calcium imaging and optogenetic data, there are motor-related Vglut2+neurons in the VLM. Unlike transient side effect of optogenetic activation, direct chemogenetic activation of VLM Vglut2+ neurons could have long-lasting motor-related side effect, preventing animals from falling asleep.

- In figure 3, the chemogenetic inhibition experiment has two strong outliers, which surely enhance the mean values. They should provide a meaningful explanation why these two animals are so different from the other 5 animals in the group. The authors should also show the profiles of hourly sleep/wake amounts during the recording time for the chemogenetic experiments shown in figures 3 and 4.

We have shown the profiles of hourly sleep/wake amounts in both chemogenetic experiments (revised fig 4 and fig5). In addition, we have provided quantification of transduced cells to explain the variation in chemogenetic inhibition experiments (supplementary fig. 8a). Those two mice (M1 and M4) had relatively higher numbers of transduced cells. Also, we have added two more mice to the revised fig 4.

- The dose of 1 mg/kg used for all chemogenetic experiments appears to be arbitrary without explanation. Since this is a pharmacologic experiment, the responses to different CNO doses should be evaluated.

As requested, we have added a lower dose of CNO (0.5mg/kg) in Supplementary fig. 8d. In chemogenetic inhibition, we observed a similar trend, but not significant change of sleep patterns after CNO treatments. Similarly, we also tried 0.5mg/kg CNO in chemogenetic activation, no significance was observed.

- The authors speculate in lines 215-221 that viral spread and non-specific optogenetic stimulation of neurons in the lateral paragigantocellular nucleus induces locomotor activity suppressing sleep during stimulation. I think this hypothesis can easily be tested by retrogradely labeling VLM neurons with channelrhodopsin, for example, from the VLPO to avoid labeling of glutamatergic neurons in the lateral paragigantocellular nucleus.

Thank you for the suggestion. We have used retrograde labeling from VLPO and performed optogenetic stimulation. The result showed that optogenetic activation of retrogradely labeled VLM neurons induced NREM sleep in awake animals following the onset of laser stimulation (Supplementary Fig 11b).

- Do the glutamatergic neurons regulate wake-sleep transition due to glutamate transmission or do these neurons co-release other transmitters?

Our slice recording data showed that VLM glutamatergic neurons activate downstream GABAergic neurons in the POA via glutamate transmission (Supplementary fig. 7). We tried to identify other potential transmitters by performing FISH and retrograde labeling. We have tried several neuropeptides, including CRH, CCK, and Carpt, based on Allen Brain database. So far, we haven't found reliable candidates for further functional characterization. One follow-up experiment is to use scRNA-seq to identify genetic markers in these sleep-related VLM cells, which will provide valuable information for co-transmitters.

Minor comments:

- In figure 1d the *Slc32a1* staining is hard to see.

The region happens to have little *slc32a1* expression, as shown in Supplementary fig. 2.

Reviewer #3 (Remarks to the Author):

- In their manuscript, the authors first combined sleep deprivation using a rod moving in the cage for 30s every 90s during 6 hours. Then they show that more *cFos+* cells are localized in a region lateral to the LPGi during sleep deprivation compared to control and sleep recovery. From the Figure, a fragmentation of sleep is visible but I don't see the quantification of each states (NREM, REM and W) quantities. Without such information it is not possible to prove that there is a sleep deprivation and recovery. It is required that the authors provide such analysis with statistical analysis of the quantities.

Per request, we have analyzed sleep patterns in sleep deprived and control mice (Supplementary fig. 1).

- Statistical analysis of the number of *cFos+* neurons is also necessary.

We have added *cFos* quantification in fig. 1c.

-The labeling of *slc32a1* is quite surprising since it does not show the presence of neurons and it is well known that this area contain a large number of GABAergic neurons.

We have added Supplementary fig. 2 to clarify this issue. The region shown in fig. 1d happens to have little expression of *slc32a1*. We observed *slc32a1* labeling in other regions from the same brain section. This expression pattern is consistent with *slc32a1* ISH data in the Allen Brain Atlas.

-In addition, the area they are showing is corresponding to the LPGi not a region lateral to it (line 84). Line 101, A1 and C1 as part of the LPGi not outside as stated.

We thank the reviewer for the clarification. We have corrected it.

-In summary, it is required that the authors make a complete sleep deprivation and recovery and demonstrate that activation of the cells is increased during NREM and not during sleep deprivation. Indeed, activation during sleep deprivation would rather mean the cells are Wake-active.

As mentioned above, we have used microendoscopic calcium imaging to examine in vivo activity of VLM glutamatergic neurons during wake and sleep states, showing sleep-active neurons in the VLM.

-The authors then use advanced specific tract tracing methodology to demonstrate that glutamatergic neurons of the LPGi excite GABAergic neurons of the POA. Overall, the results seem quite convincing but I'm lacking number of mice used in the text.

The number of mice has been added in the figure legend.

-Figure 2b is not clearly showing where are located the neurons. A delimitation of the structures is needed there. The same in C, what bregma levels it correspond to?

Information has been added to the figure and legend.

-Figure 3e, neurons retrogradely labeled are also not well localized.

The panel has been revised to show the location of the retrogradely labeled cells.

-Line 130, the fiber photometry experiment is not fully convincing since stimulation of glutamatergic neurons of the LPGi might induce NREM by another projection than the one to the POA. In such case, GABAergic neurons in the POA would also be activated.

To address this question, we have performed slice recording in POA GABAergic neurons while optogenetically stimulating ChR2-expressing VLM glutamatergic terminals (Supplementary Fig. 7). Our data demonstrated the monosynaptic excitation.

-In the following part of the paper, the authors inactivated using DREADS the POA-projecting LPGi neurons and studied its effect on sleep. They obtained a small decrease of NREM and REM and an increase in W. They concluded that « POA-projecting VLM neurons are required for the transitions from wakefulness to NREM sleep ». They cannot write such sentence since there are still many transitions. Further, it is rather an increase in NREM than an effect on transitions.

We have toned down the conclusion.

-They display Fig. 3 cumulative time, it is not adapted and they should show hour by hour values. Although an effect is seen it does not seem very strong. What is the percentage of decrease?

As requested, hourly data have been shown. There is about 35% decrease in NREM sleep over the first 3h of CNO treatment.

-I have the same requests and concerns for the stimulation experiments.

Hourly data have been shown.

-Then, the authors stimulate the glutamatergic neurons of the LPGi using optogenetic. Surprisingly, in this case they obtained a waking state. Moreover, when they stimulated during NREM they obtained wake with locomotion. They explained such results by the recruitments of glutamatergic neurons involved in waking. Unfortunately, these results are against their hypothesis of a role of these neurons in inducing NREM. To overcome the problem, they stimulate the glutamate terminals in the POA. In this case, they obtained a small increase in NREM. However, each trial was not followed by an induction of NREM. The results are therefore not very convincing.

Our imaging data demonstrated the heterogeneity of VLM glutamatergic neurons, including both wake-active (mostly motor-related) and sleep-active neurons. Consistent with this, direct optogenetic stimulation of VLM glutamatergic neurons induced mixed results. Our terminal stimulation experiments as well as chemogenetic experiments demonstrated that POA-projecting glutamatergic neurons in the VLM can induce NREM sleep. The effect of terminal stimulation is smaller compared to soma stimulation. We have discussed the possibilities in the text.

REVIEWER COMMENTS

Reviewer #1 (Remarks to the Author):

The Authors have sufficiently addressed several of my previous comments. The new experiments included in this version further improve the quality of the overall manuscript. I only have the following comments.

- 1. I am still not convinced with the sleep-deprivation specifically reflecting wake-to-sleep transitions and the interpretation. The authors now included the requested calcium imaging data, which show that there are NREM-active neurons in this region. It is not clear if these neurons are active prior to NREM sleep which will support their role in sleep initiation (wake-sleep transition). Transitions to NREM sleep observed with photostimulation of POA-projecting neurons as well as VLM terminals in the POA do support this idea, but further analysis of existing photometry data to isolate the activity changes of these NREM-active neurons during transitions would provide stronger support to the conclusions of the manuscript.**
- 2. Minor: Fig 3C panel 1 – NTS is misspelled**
- 3. Suppl Fig 11 b and C could have been in the main figures as they contain crucial data sets. Also, this figure is mislabeled as supplementary Fig 11d in the text.**

Reviewer #2 (Remarks to the Author):

I have seen a previous version of the manuscript as reviewer #2. I raised concerns that the conclusions of the manuscript are pre-mature. I am impressed with the thorough revision of the manuscript, especially adding new data of microendoscopic calcium imaging that show a population of sleep-active neurons in the VLM. Many of my concerns have been resolved and I now believe in the existence of sleep-active neurons in the VLM.

However, I am still not convinced that the VLM projections to the POA are the key circuit that reliably initiates NREM sleep in awake mice, as stated in the abstract. This conclusion is only supported by data in figure 7 but based on these data it is certainly an overstatement to say the POA-projecting VLM neurons reliably induce NREM sleep. What is the effect of optogenetic VLM neurons activation when POA neurons are blocked, e.g. by inhibition/ablation?

Reviewer #3 (Remarks to the Author):

The authors did improve the manuscript notably by adding calcium imaging of the LPGi glutamatergic neurons

Their results are both new and interesting and there is a large number of data presented.

There are however still open questions who should be addressed.

First, they obtained an increase in the number of cFos cells in the LPGi after a mild sleep deprivation. Indeed, REM is nearly abolished but NREM is only slightly decreased and fragmented. Such deprivation interestingly induced cFos expression in the LPGi. The question is why? The authors interpret that such increase is due to the increase number of transitions from W to NREM. However, other factors could be involved as well. In particular, such labeling could be due to the increase in waking. It would be in line with their results showing that stimulation of glutamatergic neurons of the LPGi induces Waking. It is also fitting with their results showing that optogenetic stimulations of the LPO projecting neurons first rather induces Waking followed by an increase in NREM. Further, 57% of the LPGi glutamatergic cells are wake-active neurons while only 16% are NREM active.

I also wonder whether the LPO projecting neurons are not adrenergic and noradrenergic neurons. Indeed, it has been shown that the C1 neurons are also glutamatergic and

project to the LPO. It is therefore likely that a large proportion of the LPGi neurons projecting to the LPO are adrenergic neurons as reported before in multiple studies. The authors should make such simple verification to confirm that this is the case. This would fit also with the idea that they excite wake-active neurons since adrenergic neurons are belonging to such class. Another important control is to verify whether the glutamatergic neurons of the LPGi projecting to the LPO are cFos after the induction of waking. This is a simple experiment to perform.

It might be also that the glutamatergic neurons project to Wake-active GABAergic LPO neurons and by this means rather inhibit sleep since these neurons project locally to sleep-active GABAergic neurons of the VLPO. Then, the mechanism responsible for the delayed increase of sleep could be indirect by means of a mechanism which needs to be identified.

One way to better understand what's going on would be also to make cFos staining after chemogenetic or better optogenetic stimulations. This would give an idea on the state induced looking at Fos staining in sleep and wake inducing systems and in the cortex.

REVIEWER COMMENTS

Reviewer #1 (Remarks to the Author):

1. I am still not convinced with the sleep-deprivation specifically reflecting wake-to-sleep transitions and the interpretation. The authors now included the requested calcium imaging data, which show that there are NREM-active neurons in this region. It is not clear if these neurons are active prior to NREM sleep which will support their role in sleep initiation (wake-sleep transition). Transitions to NREM sleep observed with photostimulation of POA-projecting neurons as well as VLM terminals in the POA do support this idea, but further analysis of existing photometry data to isolate the activity changes of these NREM-active neurons during transitions would provide stronger support to the conclusions of the manuscript.

We thank the reviewer for the suggestion. We have analyzed the activity of these NREM-active neurons during the transitions between wake and NREM sleep, demonstrating that these neurons display increased activity during the wake-to-NREM transitions, but no significant changes during the NREM-to-wake transitions (new Fig. 2d).

2. Minor: Fig 3C panel 1 – NTS is misspelled

We have corrected it.

3. Suppl Fig 11 b and C could have been in the main figures as they contain crucial data sets. Also, this figure is mislabeled as supplementary Fig 11d in the text.

We have moved Suppl Fig 11b-c to the main Fig. 7. Also, we have changed the mislabeling.

Reviewer #2 (Remarks to the Author):

However, I am still not convinced that the VLM projections to the POA are the key circuit that reliably initiates NREM sleep in awake mice, as stated in the abstract. This conclusion is only supported by data in figure 7 but based on these data it is certainly an overstatement to say the POA-projecting VLM neurons reliably induce NREM sleep. What is the effect of optogenetic VLM neurons activation when POA neurons are blocked, e.g. by inhibition/ablation?

Thank you for the comments. As suggested, we have tried both ablation and inhibition experiments. For ablation, we bilaterally injected DTA (AAV-FLEX-DTA, N=5 animals) or TeNT (AAV-FLEX-TeNT, N=6 animals) in the POA of GAD2-Cre or Vgat-Cre mice (AAV-CaMKII-ChR2 also injected in VLM for optogenetic stimulation). However, almost all mice (10 out of 11 animals) died between the period of surgery and experiments. We speculated that this could be related to the loss/alteration of thermal regulation as the preoptic area is greatly involved in this process. We then perform pharmacological inhibition experiments. By local infusion of glutamate receptor antagonist NBQX in the POA, we have shown that POA inhibition suppressed optogenetic-induced wake-to-NREM transitions and subsequent NREM sleep (Supplementary Fig. 15). Also, we have slightly toned down the conclusion in the abstract.

Reviewer #3 (Remarks to the Author):

First, they obtained an increase in the number of cFos cells in the LPGi after a mild sleep deprivation. Indeed, REM is nearly abolished but NREM is only slightly decreased and fragmented. Such deprivation interestingly induced cFos expression in the LPGi. The question is why? The authors interpret that such increase is due to the increase number of transitions from W to NREM. However, other factors could be involved as well. In particular, such labeling could be

due to the increase in waking. It would be in line with their results showing that stimulation of glutamatergic neurons of the LPGi induces Waking. It is also fitting with their results showing that optogenetic stimulations of the LPO projecting neurons first rather induces Waking followed by an increase in NREM. Further, 57% of the LPGi glutamatergic cells are wake-active neurons while only 16% are NREM active.

We agree that there could be other interpretations for c-Fos data. So, we did microendoscopic calcium imaging. By adding new analysis of activity during the wake-sleep transitions, we have shown that these NREM active neurons display increased activity in the wake-to-NREM transitions, but not NREM-to-wake transitions (Fig. 2d). Based on our imaging and optogenetic data, we agree that there are wake-related glutamatergic neurons in the VLM. However, among VLM glutamatergic neurons, we show that POA-projecting neurons are NREM-promoting neurons (see new Fig 6, Fig 7g-h, Fig 8, and supplementary Fig 12b).

I also wonder whether the LPO projecting neurons are not adrenergic and noradrenergic neurons. Indeed, it has been shown that the C1 neurons are also glutamatergic and project to the LPO. It is therefore likely that a large proportion of the LPGi neurons projecting to the LPO are adrenergic neurons as reported before in multiple studies. The authors should make such simple verification to confirm that this is the case. This would fit also with the idea that they excite wake-active neurons since adrenergic neurons are belonging to such class.

Thank you for the comment. We have performed TH staining in retrogradely labeled mice. Our data show that only ~18% retrogradely labeled VLM cells are TH+ (Supplementary Fig. 5b).

Another important control is to verify whether the glutamatergic neurons of the LPGi projecting to the LPO are cFos after the induction of waking. This is a simple experiment to perform.

Even though activation of these VLM vglut2+ neurons induce brief wake in sleeping mice, it often followed by NREM sleep (Supplementary Fig. 12a). Given the temporal resolution of cFos staining, we think that it's difficult to capture meaningful cFos signals to delineate the dynamic sleep/wake process. See below for more information.

It might be also that the glutamatergic neurons project to Wake-active GABAergic LPO neurons and by this means rather inhibit sleep since these neurons project locally to sleep-active GABAergic neurons of the VLPO. Then, the mechanism responsible for the delayed increase of sleep could be indirect by means of a mechanism which needs to be identified. One way to better understand what's going on would be also to make cFos staining after chemogenetic or better optogenetic stimulations. This would give an idea on the state induced looking at Fos staining in sleep and wake inducing systems and in the cortex.

As suggested, we have performed cFos staining after optogenetic stimulation. As expected, we found increased cFos signals in VLM and some direct downstream targets, such as POA and vIPAG (Supplementary Fig 14). Among wake-inducing systems, we observed increased cFos in the Locus coeruleus (LC). We speculate that this might be caused by projection from wake-related neurons in the VLM. Indeed, previous studies show that C1 cells (likely partially labeled by AAV-DIO-ChR2 injection in Vglut2-Cre mice) project to LC. In the cortex, we didn't see significant changes.

REVIEWER COMMENTS

Reviewer #1 (Remarks to the Author):

The authors have adequately addressed my previous comments. I do not have any further comments. I wish to congratulate the authors for producing this nice work.

Reviewer #2 (Remarks to the Author):

I appreciate the author's efforts to perform additional pharmacological experiments to clarify the extent to which glutamatergic medulla projections to the preoptic area contribute to optogenetically induced NREM sleep (Suppl. Fig. 15). As expected, the data may suggest that glutamatergic medulla neurons projecting to other brain areas are also involved in the induction of NREM sleep. I have no further comments.

Reviewer #3 (Remarks to the Author):

The revision provided brings some welcome additional experiments requested. I have still some additional concerns on the paper.

First, the title is misleading. It should state that Ventrolateral medullary glutamatergic neurons projecting to the preoptic area promote NREM sleep. Indeed, the authors don't demonstrate a role in transitions. Further, most glutamatergic neurons in the VLM are wake-active and wake inducing. Only a very small proportion of cells are NREM active (16%). Many more cells are wake-active (57%) or REM-active (27%). It indicates that the involvement of the VLM glutamatergic neurons in NREM sleep is very limited compared to its role in W or REM sleep. In that regard, it would have been in a way more logical to look at the role of these NREM cells in the inactivation of the W and REM-active local neurons rather than to focus on a long range projection to the preoptic area.

The authors need to discuss that point and to make an hypothesis on the mechanisms responsible for the activation of the glutamatergic NREM active neurons and their inhibition during the other states. What could be the role of such neurons in NREM sleep generation? What position they would have in the network? A diagram would help.

Concerning cFos staining in Fig. 1 after sleep deprivation. I still don't get the meaning of the results obtained. They should provide the quantity of sleep in that figure. When looking to SupFig 1, it shows a decrease of NREM and more over a complete disappearance of REM sleep and an increase in W. Therefore, their staining reflect first neurons activated during W rather than during NREM. The fragmentation induced is also not huge since they leave 90s between each 30s rod movement. In fact, the neurons activated during NREM should be cFos labeled after recovery sleep. It does not seem to be the case? The neurons activated during W should correspond to W-activated neurons. In summary, the experimental design of this first part is not good and the authors should have made complete NREM deprivation and rebound.

Interestingly 18% of the VLM neurons projecting to the preoptic area are noradrenergic. I wonder whether these neurons are cFos positive during deprivation?

In their « in vitro » experiments on the projection from the VLM to the POA showing excitatory projections to GABAergic neurons, they would need to show that norepinephrine is also inhibiting these neurons to demonstrate that they are sleep-active. Indeed, there is a subpopulation of Wake-active neurons in the POA which are excited by norepinephrine.

The results obtained with chemogenetic inhibition shown in Fig. 5 are quite convincingly showing a role for the VLM glutamatergic neurons projecting to the preoptic area in NREM sleep promotion.

The results obtained by optogenetic experiments and shown in Fig. 7 are on the contrary showing a role of the glutamatergic neurons of the VLM in wakefulness. These results should be put in the supplementary material since they are not in line with the

chemogenetic ones. They are rather showing that most of the glutamatergic neurons of the region are inducing wakefulness as stated above.

In summary, overall the demonstration of a role of VLM glutamatergic neurons in NREM projecting to the preoptic area is new and interesting. Some results are however not well organized and the cFos experiments should be better implemented. Finally, what place have these neurons in the NREM network should be addressed.

Response to reviewer comments

Reviewer #1 (Remarks to the Author):

The authors have adequately addressed my previous comments. I do not have any further comments. I wish to congratulate the authors for producing this nice work.

We thank the reviewer for suggested experiments and analyses to improve the quality of our work.

Reviewer #2 (Remarks to the Author):

I appreciate the author's efforts to perform additional pharmacological experiments to clarify the extent to which glutamatergic medulla projections to the preoptic area contribute to optogenetically induced NREM sleep (Suppl. Fig. 15). As expected, the data may suggest that glutamatergic medulla neurons projecting to other brain areas are also involved in the induction of NREM sleep. I have no further comments.

We thank the reviewer for thoughtful suggestions to this work.

Reviewer #3 (Remarks to the Author):

The revision provided brings some welcome additional experiments requested. I have still some additional concerns on the paper. First, the title is misleading. It should state that Ventrolateral medullary glutamatergic neurons projecting to the preoptic area promote NREM sleep. Indeed, the authors don't demonstrate a role in transitions.

We agree that including the preoptic area in the title would be more informative. We suggest the title: Control of Non-REM sleep by ventrolateral medulla glutamatergic neurons projecting to the preoptic area.

Further, most glutamatergic neurons in the VLM are wake-active and wake inducing. Only a very small proportion of cells are NREM active (16%). Many more cells are wake-active (57%) or REM-active (27%). It indicates that the involvement of the VLM glutamatergic neurons in NREM sleep is very limited compared to its role in W or REM sleep. In that regard, it would have been in a way more logical to look at the role of these NREM cells in the inactivation of the W and REM-active local neurons rather than to focus on a long range projection to the preoptic area.

It would be nice to study the role of NREM-active VLM neuron while inactivating wake- and REM-active neurons. However, there is no feasible way to specifically target those wake- and REM-active glutamatergic neurons without affecting NREM-active neurons until specific marker genes (and mouse lines) are identified for each cell population. We plan to search for these marker genes in future studies.

The authors need to discuss that point and to make an hypothesis on the mechanisms responsible for the activation of the glutamatergic NREM active neurons and their inhibition during the other states. What could be the role of such neurons in NREM sleep generation? What position they would have in the network? A diagram would help.

We thank the reviewer for suggestions. Based on their activity during the Wake-NREM transitions (Fig. 2d) and their long-lasting effect on inducing NREM sleep with optogenetic

activation (Fig. 7c and Fig. 7g), we hypothesize that these glutamatergic VLM neurons play an initiation role in NREM sleep generation. We have added a diagram in the supplementary Fig. 13c to illustrate possible connections between the VLM neurons and other NREM-related brain regions in the network. Regarding the mechanisms responsible for their activation during NREM sleep and inhibition during the other states, unfortunately, we have little understanding so far. It requires anatomic and functional characterization of the input neurons to these VLM neurons. As mentioned, VLM glutamatergic neurons are heterogeneous, including wake-active, REM-active, and NREM-active neurons. Retrograde tracing from VLM could identify input neurons, but it will take a lot of effort to dissect which neurons (among many wake/sleep-related brain areas) actually project to NREM-active VLM neurons, and whether their transmission is excitatory or inhibitory. We think the work is beyond this study. We will perform these experiments in a follow-up study.

Concerning cFos staining in Fig. 1 after sleep deprivation. I still don't get the meaning of the results obtained. They should provide the quantity of sleep in that figure. When looking to SupFig 1, it shows a decrease of NREM and more over a complete disappearance of REM sleep and an increase in W. Therefore, their staining reflect first neurons activated during W rather than during NREM. The fragmentation induced is also not huge since they leave 90s between each 30s rod movement. In fact, the neurons activated during NREM should be cFos labeled after recovery sleep. It does not seem to be the case? The neurons activated during W should correspond to W-activated neurons. In summary, the experimental design of this first part is not good and the authors should have made complete NREM deprivation and rebound.

The purpose of our cFos experiments is to identify candidate neuronal populations that are likely involved in wake-sleep transitions. Thus, we designed a SD paradigm with frequent transitions. We argue that c-Fos+ cells reflect the increased wake-sleep transitions, not wakefulness. The control group, having the most amount of wake time (supFig 1C), actually showed less c-Fos+ cells (Fig. 1C). Similarly, the RS (rebound) group had more total NREM sleep time (but less transitions, supFig 1e), also showed relatively lower c-Fos+ cells, compared to SD group. As mentioned earlier, we agree that there are other interpretations regarding the nature of c-Fos signals. So, we followed up with in vivo imaging, which provides more information of neuronal activity during wake/sleep cycles, particularly to distinguish wake-to-NREM transitions from NREM-to-wake transitions (Fig. 2d).

Interestingly 18% of the VLM neurons projecting to the preoptic area are noradrenergic. I wonder whether these neurons are cFos positive during deprivation?

To address this question, we performed co-staining experiments of cFos and TH. Our data show almost no overlap (3.85%) between TH and cFos in the VLM region (Supplementary Fig. 5c).

In their « in vitro » experiments on the projection from the VLM to the POA showing excitatory projections to GABAergic neurons, they would need to show that norepinephrine is also inhibiting these neurons to demonstrate that they are sleep-active. Indeed, there is a subpopulation of Wake-active neurons in the POA which are excited by norepinephrine.

As suggested, we tried slice recording to characterizing the potential norepinephrine transmission between VLM TH+ neurons and POA GABAergic neurons. In TH-Cre mice, we bilaterally injected AAV-DIO-ChR2 in the VLM and bilaterally injected AAV-mDlx-mRuby2 in the POA. Then, we sliced the brains and performed patch-clamp recording in the Ruby+ cells in the POA while optogenetically activating the TH+ VLM terminals. We tried both EPSC (excitatory,

recorded 17 cells) and IPSC (inhibitory, recorded 18 cells) protocols from 4 mice, but didn't observe any optogenetic-evoked response. We speculated that a small amount of VLM TH+ neurons have sparse projections in the POA, which makes it difficult to find post-synaptic neurons. This result (not included in supplementary figures) along with fos co-staining data might also imply a limited role of VLM TH+ neurons in our manipulation experiments.

The results obtained with chemogenetic inhibition shown in Fig. 5 are quite convincingly showing a role for the VLM glutamatergic neurons projecting to the preoptic area in NREM sleep promotion.

The results obtained by optogenetic experiments and shown in Fig. 7 are on the contrary showing a role of the glutamatergic neurons of the VLM in wakefulness. These results should be put in the supplementary material since they are not in line with the chemogenetic ones. They are rather showing that most of the glutamatergic neurons of the region are inducing wakefulness as stated above.

We would like to keep optogenetic activation in the main figures. We agree that some stimulation experiments (Fig. 7a-e) involve mixed populations, including wake-active cells. However, to our knowledge, this is the first optogenetic study demonstrating that ChR2 activation of a group of neurons can induce long-lasting natural-like NREM sleep with such a high probability, following the offset (regrettably not onset) of laser stimulation. In previous optogenetic studies, ChR2 activation usually induces a short period of NREM sleep, and the effect quickly disappears when the laser is off. Although activation of mixed populations lead to complicated wake-then-sleep phenomenon, nevertheless, this study presents striking results of NREM control.

In summary, overall the demonstration of a role of VLM glutamatergic neurons in NREM projecting to the preoptic area is new and interesting. Some results are however not well organized and the cFos experiments should be better implemented. Finally, what place have these neurons in the NREM network should be addressed.

We thank the reviewer for suggestions and comments to elevate the quality of this study. As mentioned above, we have discussed the role of these VLM glutamatergic neurons in regulating NREM sleep.

REVIEWERS' COMMENTS

Reviewer #3 (Remarks to the Author):

The authors answered to my request properly and the manuscript is I believe ready for publication.

RESPONSE TO REVIEWERS' COMMENTS

Reviewer #3 (Remarks to the Author):

The authors answered to my request properly and the manuscript is I believe ready for publication.

We are thankful to the reviewer for constructive suggestions and comments to improve the quality of this work.